# communications
# engineering

# Energy efficient perching and takeoff of a miniature rotorcraft

Yi-Hsuan Hsiao [1,2,5], Songnan Bai [2,5], Yongsen Zhou[3,5], Huaiyuan Jia[2,5], Runze Ding [2], Yufeng Chen[1], Zuankai Wang [4✉] & Pakpong Chirarattananon [2,3✉]

The flight time of aircraft rapidly decreases with smaller scales because the lift-to-drag ratio decreases when scaling down. Aerial-surface locomotion, or perching is one energy efficient solution to prolong the fight time by maintaining the drone at a high vantage point. Current perching strategies require additional components to ensure robots firmly attach to the surfaces, and able to detach afterwards, resulting in increased power consumption owing to the added weight. Here, we report a 32-g rotorcraft with the ability to repeatedly perch and take off from overhangs and walls on different wet and dry substances. A propelling thrust is used to support the robot to keep rotorcraft balance against the surface. Integrating with the mussel-inspired wet adhesives, the rotorcraft dispenses the additional components required for attachment and taking off. The final rotorcraft is 32.15 g, only 1.09 g heavier than the original prototype, but shows a 50% and 85% reduction in power consumption when perching on ceilings and walls respectively. The saved power leads to a fourfold increase in the total mission time.

[1] Department of Electrical Engineering and Computer Science, Massachusetts Institute of Technology, Cambridge, MA, USA. [2] Department of Biomedical Engineering, City University of Hong Kong, Tat Chee Avenue, Kowloon Tong, Hong Kong SAR, China. [3] Department of Mechanical Engineering, City University of Hong Kong, Tat Chee Avenue, Kowloon Tong, Hong Kong SAR, China. [4] Department of Mechanical Engineering, The Hong Kong Polytechnic University, Hung Hom, Kowloon, Hong Kong SAR, China. [5] These authors contributed equally: Yi-Hsuan Hsiao, Songnan Bai, Yongsen Zhou, Huaiyuan Jia. ✉email: zk.wang@polyu.edu.hk; pakpong.c@cityu.edu.hk

Power is one of the dominant considerations for flight. At low Reynolds numbers, the energetic requirement for flight is aggravated by strengthened viscous forces[1]. Scaling down an aircraft results in a decreased lift-to-drag ratio. The inferior aerodynamic efficiency renders flight time of small drones acutely restricted, rapidly diminishing to a couple of minutes for sub-100-g vehicles[2].

Over the past several years, tremendous efforts have been spent in improving energetic efficiency of robotic flight at small scales[3–8]. Besides, multiple workaround solutions have been explored, including the usage of alternative energy sources[9–12]. Furthermore, multimodal operation—a bioinspired approach, such as hybrid aerial and terrestrial locomotion has been experimentally shown to substantially expand the working range provided suitable operating conditions[13–16].

Aerial-surface locomotion, or perching, emerges as a promising avenue that allows aerial vehicles to maintain a high vantage point for a prolonged period with less power consumption[17]. Among existing small flying robots with the ability to perch (see Supplementary Fig. 1 and Supplementary Table 1), actuated grippers are the most common mechanisms that enable the robots to grab onto branches[18–30]. Relatively few vehicles are able to land and take off from walls[31–37] and ceilings[37–39]. To establish a firm contact with flat surfaces, electroadhesion[38,39], gecko-inspired dry adhesives[33,35] and small needles, or microspines[31,32,34,37], have been employed. With relatively weak adhesion pressure (less than 1 kPa), the use of electrostatic forces results in disproportionately large adhesive pads and is still limited to robots under 20 g for perching on dry surfaces[38,39]. For dry pressure-sensitive adhesives (PSAs) and spines[31,32,34,35,37], they are limited to smooth (glass[35]) and rough surfaces (wood[31] and concrete[32,34]), respectively. Moreover, both adhesives were deployed with a servo or motor and suitable mechanisms (such as preloaded springs) to ensure the robots can detach afterward (with an exception for the robot in ref. [22], which directly anchors on a branch when perching). These additional components account for an appreciable portion of the final vehicle mass. As illustrated in Supplementary Fig. 1, the perching mechanisms (including added actuators) constitute over 15% of the total mass for vehicles under 100 g. While added mass generally poses less of a challenge for terrestrial locomotion, the aerial domain's reliance on generating sufficient lift and the strong dependence of power on mass renders the mass budget for flying vehicles, especially small drones, far more stringent. For multirotor platforms, momentum theory predicts the scaling between the aerodynamic power thrust $T$ of a spinning propeller as $P_a \sim T^{3/2}$[40,41], implying that a 15% increase in weight, for instance, nominally leads to a 23% rise in power consumption. In other words, the introduced capability to perch and conserve energy simultaneously compromises flight endurance substantially.

This work tackles the major shortcoming of existing perching methods for Micro Aerial Vehicles (MAVs) to rest on both an overhang or a wall. The proposed strategy, which combines the airflow-surface interactions[41] with mussel-inspired wet adhesive[42–45], dispenses the need for additional actuators for engaging and disengaging the mechanisms. When incorporated in to a miniature rotorcraft, the final mass of the robot is only 3% heavier than the original prototype (32.15 g versus 31.06 g, see Supplementary Fig. 1).

To facilitate repeatable perching maneuvers without extra actuators, the adhesive pads are incorporated into a lightweight customized airframe with a passive mechanism, and the use of the proximity effect is integral. The developed framework differs from previous implementations in two aspects. First, to preload the wet adhesive, the propelling thrust is used directly (instead of the use of elastic energy stored in a mechanism[15,18–22,24,31,32,34,37]). This is feasible as the robot takes advantage of the aerodynamic effect induced by a nearby surface. The proximity effect[41,46–48], akin to the well-known ground effect[47,49], amplifies the propelling thrust by over a factor of two. Second, when the robot is attached to the surface, it is supported by both the adhesion force and small propelling thrust to stay in force and moment equilibrium. Despite the need for small thrust while perching, the power consumption is immensely reduced as the aerodynamic efficiency is notably boosted by the proximity effect. When instructed, further lowering or removing the thrust commands allows the vehicle to seamlessly detach from the surface by peeling off the adhesive. Therefore, thrust assistance replaces the need for extra actuation or a sophisticated mechanism, rendering it suitable for small vehicles with limited payload. The strategy is compatible with both walls and ceilings. The developed palm-sized quadcopter, as a result, is able to substantially extend its mission time without compromising on flight endurance.

While the hybrid perching strategy is compatible with different types of adhesion methods, the choice primarily depends on anticipated environments. Microspines need surfaces with prominent rugosity. The mussel-inspired wet adhesive offers certain advantages. Unlike dry adhesive pads constructed with hair-like microstructures, of which van der Waals adhesion rapidly deteriorates when dampened[50] (the susceptibility to water also applies to electrostatic adhesion presented in refs. [38,39,51]), the main composition of the biomimetic polymeric adhesive film in this work is based on multi-proteins secreted by marine mussels. Previous studies have identified 3,4-dihydroxyphenyl-L-alanine (DOPA) as the chemical basis for the development of mussel-mimetic polymer[52,53]. By incorporating the DOPA moiety into the matrices of PSA, DOPA-modified adhesives demonstrate repeatable attachment to a variety of surfaces in both dry and wet environments[42,44,45]. Moreover, compared to other recent state-of-the-art approaches compatible with wet surfaces[26,54,55], the DOPA-based polymeric adhesive benefits from the wide range of workable surfaces (both smooth and rough, unlike commonly used dry adhesive and spines that are restricted to polished and uneven substrates, respectively) and the feasibility to be used passively.

The proposed mechanism was analyzed to obtain the conditions required for the 32-g robot to attach and take off from horizontal and vertical surfaces. Taking into account the contribution of the propelling thrust and the anticipated proximity effect, the biomimetic adhesive was experimentally characterized and verified for its adhesion pressure and reusability. Based on the devised maneuvers, extensive flight experiments were conducted to demonstrate the robot repeatedly perching on different dry and wet substrates (Supplementary Movie 1). The benefits of the hybrid mechanism are reflected in the marked decrease in power consumption and improvement on operational endurance, as well as the ability to perform the maneuvers with only onboard feedback. Furthermore, the enhancement was accomplished with negligible impact on the flight times thanks to the minimal added weight.

## Results

**Hybrid strategy for perching on horizontal and vertical surfaces.** Focusing on the severely constrained payload of small aerial vehicles, the proposed perching strategy integrates two distinct mechanisms: proximity effects and biomimetic wet adhesion, to assist the palm-sized robot to perch and takeoff from both horizontal and vertical surfaces without the need for additional actuators or elaborate structures (Supplementary Movie 1). This enables the vehicle to rest and expend appreciably less power compared to flying.

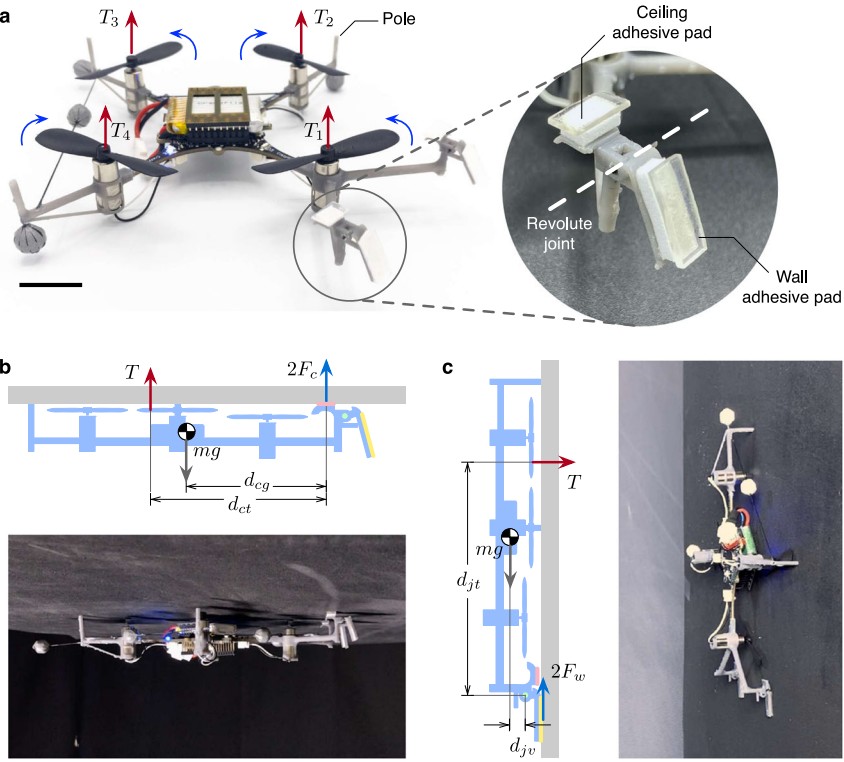

**Fig. 1 Overview of the developed prototype and the perching strategy. a** Photograph of the 32-g robot showing two pairs of adhesive pads and pole structures. The wall adhesive pads reside on passive joints. The rotor thrusts are labeled $T_1$, $T_2$, $T_3$, and $T_4$. The scale bar is 2 cm. **b** A diagram and a photo showing forces on the robot when perched on an overhang (normal forces are omitted). The weight of the robot ($mg$) is supported by the collective thrust ($T$) and the adhesive ($2F_c$) **c** A diagram and a photo showing forces on the robot when perched on a vertical surface (normal forces are omitted). The weight of the robot is entirely supported by the adhesive ($2F_w$).

The developed prototype incorporates two pairs of wet adhesive pads: ceiling pads and wall pads, and three rigid poles located on the outer perimeter of the vehicle for preventing the propellers from colliding with a surface (Fig. 1a). Each pair of the bio-inspired adhesive pads, decorated on PDMS bases, are separately engaged when the robot perches on a ceiling (Fig. 1b) or a wall (Fig. 1c). To remain stationary, the required collective thrust, $T = \sum_{i=1}^{i=4} T_i$ (Fig. 1a), is radically reduced from the nominal flight condition, $T = mg$, thanks to the presence of the adhesion force ($2F_c$ from the ceiling adhesive in Fig. 1b or $2F_w$ from the wall adhesive in Fig. 1b). Meanwhile, the vicinity of the surface to the propellers also substantially decreases aerodynamic power and, subsequently, the power consumption of the motors as reported in ref. [41]. The combined effects dramaticaly expand the operational time of the vehicle.

For the robot to rest underneath an overhang, the static equilibrium conditions must be met. Without modeling normal forces acting on the poles explicitly, the condition for the translational dynamics is captured by (see Fig. 1b)

$$2F_c + \sum_{i=1}^{i=4} T_i = 2F_c + T \geq mg, \qquad (1)$$

in which the ceiling adhesive pads lower the required collective thrust, contributing to power conservation. The static condition for the rotational dynamics evaluated at the contact point of the adhesive pads necessitates

$$Td_{ct} \geq mgd_{cg}, \qquad (2)$$

where $d_{ct}$ and $d_{cg}$ represent effective moment arms of the collective thrust force and the robot's weight with respect to the attachment point (Fig. 1b). The inequality arises from the exclusion of the normal forces at the poles. Together, Equations (1) and (2) suggest

that there exists an equilibrium state with $T < mg$ as long as $F_c > 0$ and $d_{ct} > d_{cg}$. The decrease in the required collective thrust is dictated by the magnitude of the adhesion force and the mechanical advantage $d_{ct}/d_{cg}$.

When it comes to the wall perching, a similar analysis applies, but the wall adhesive pads are anchored to the airframe via the revolute joints (Fig. 1a). When perched, the pads are assumed adhered to the wall and the weight is balanced by the shear adhesion $F_w$ produced by the adhesive (Fig. 1c):

$$2F_w = mg, \qquad (3)$$

which is independent of $T$. Simultaneously, the equilibrium state of the rotational dynamics of the main robot body (excluding the wall adhesive pads, see Fig. 1c), is evaluated about the joint axes. This is achieved when

$$Td_{jt} \geq mgd_{jv}, \qquad (4)$$

in which $d_{jt}$ is the effective moment arm of the collective thrust force with respect to the joint axes and $d_{jv}$ is the distance between the center of gravity and the joint axes (Fig. 1c). Similarly, the inequality stems from the omission of the normal forces. Likewise, the outcome points to the decrease in the required collective thrust inversely proportional to the mechanical advantage $d_{jt}/d_{jv}$.

In both scenarios, as the propellers are close to surfaces, the proximity effect increases the aerodynamic efficiency of the propellers. When the input power or motor command is maintained, this leads to an increase in the propelling thrust. Alternatively, the introduction of a nearby surface lowers the power consumption of a spinning propeller when the thrust force is kept constant.

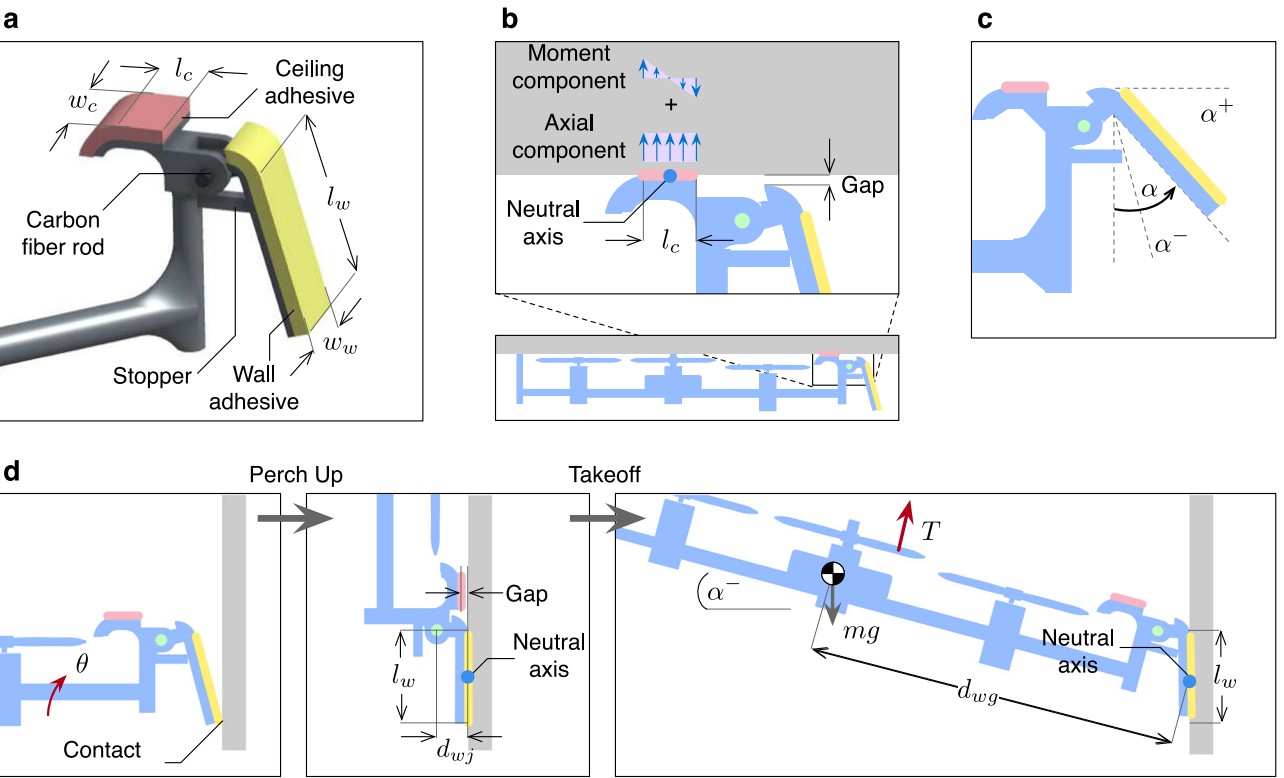

**Fig. 2 Design of the hybrid perching mechanism.** The variables $l_c$, $l_w$, $w_c$, $w_w$, $d_{wj}$, and $d_{wg}$ are length parameters. **a** Closed-up view of the attachment mechanism consisting of a ceiling adhesive pad and a wall adhesive pad with a revolute joint. **b** Local distribution of the adhesion force over the ceiling adhesive pad. **c** Joint limits ($\alpha^+$, $\alpha^-$) of the passive hinge as prescribed by the angle $\alpha$. **d** Schematic diagrams illustrating the process and major requirements of the wall perching maneuver as dictated by the weight $mg$ and collective thrust $T$.

**Mechanical design and adhesion requirements for detachable perching**. The augmentation of aerodynamic forces from surfaces and adhesion enables the rotorcraft to dissipate less power while staying elevated. The implementation challenge lies in the strategy that lets the robot to attach and detach from a wide variety of horizontal and vertical surfaces, dry and wet, without additional actuators or hefty components. A lightweight mechanism was tailored for perching with compatibility with a large range of adhesion pressure anticipated for different surface types.

To perch on both overhangs and walls, the pairs of ceiling and wall adhesive pads are separated (Figs. 1a and 2a). The wet adhesive is coated on the PDMS layers and installed on the 3D-printed bases. The ceiling adhesive pads constitute a rigid extension of the airframe, whereas the wall pads are free to rotate in one degree of freedom. This allows the wall adhesive pads to freely switch between two configurations: upward and downward, preventing the wall pads from interfering during the ceiling perching.

While perching on an overhang (Fig. 2b), the wall adhesive pads remain in the nominal downward configuration, leaving a small gap between the adhesive and the surface. The ceiling pads make full contact with the surface, providing the tensile force to handle part of the robot's weight. With the assisted thrust, there exists no minimum limit of the adhesion force required for the robot to stay perched on a ceiling. The adhesion can be expressed as $F_c = \sigma_a A_c$, where $\sigma_a$ represents the tensile stress and $A_c = l_c w_c$ stands for the surface area of the ceiling adhesive (Fig. 2a and Supplementary Table 2). To preload the adhesive pads to the ceiling, a large propelling thrust, amplified with the proximity effect, is applied. Compared to a single pad setting, the pair configuration stabilizes the possible yaw motion of the vehicle via shear adhesion.

To facilitate the unactuated detachment, the ceiling adhesive pads are displaced from the center of mass by the distance $d_{cg}$ (Fig. 1b and Supplementary Table 2). This guarantees the robot is able to take off from multiple surfaces by peeling off the adhesive, demoting the importance of the surface-dependent maximum adhesion pressure of the adhesive. To detach, the vehicle is commanded to briefly wind down or stop all the propellers ($T \rightarrow 0$). The condition for the robot to remain perched is the balance of force and moment. This induces non-uniform stress across the adhesive pads. As detailed in Supplementary Note 1 and Supplementary Fig. 2, the strategic placement of the ceiling adhesive pads amplifies the maximum tensile pressure the adhesive pads undergo in the peeling-off process by approximately 60 times compared to the case where the pads were put directly on top of the center of mass. As a consequence, the takeoff condition is relatively insensitive to the surface properties, applied preload, and the size of the adhesive pads.

When it comes to the wall perching, the incorporation of the passive joints (Fig. 2c) serves two purposes: precluding the ceiling adhesive from sticking to the wall (and vice versa) and assisting the perch up motion towards the wall. Nominally, the ceiling adhesive pads are situated on top of the robots, allowing them to readily adhere to the ceiling. With the passive joints, the wall adhesive pads are oriented almost vertically when the robot is flying (see Fig. 2d), permitting them to be in touch with a wall upon approaching. With the revolute joints, the initial contact assists the perch-up maneuver and the designated gap prevents the ceiling adhesive pads from adhering to the wall. The mechanical features, acting as joint stoppers (Fig. 2c), restrict the angle between the pads and the vertical axis to the predefined limits $\alpha^- < \alpha < \alpha^+$ (Fig. 2c). The joint limits facilitate the approach, perching, and peeling-off process (Fig. 2d). The shear

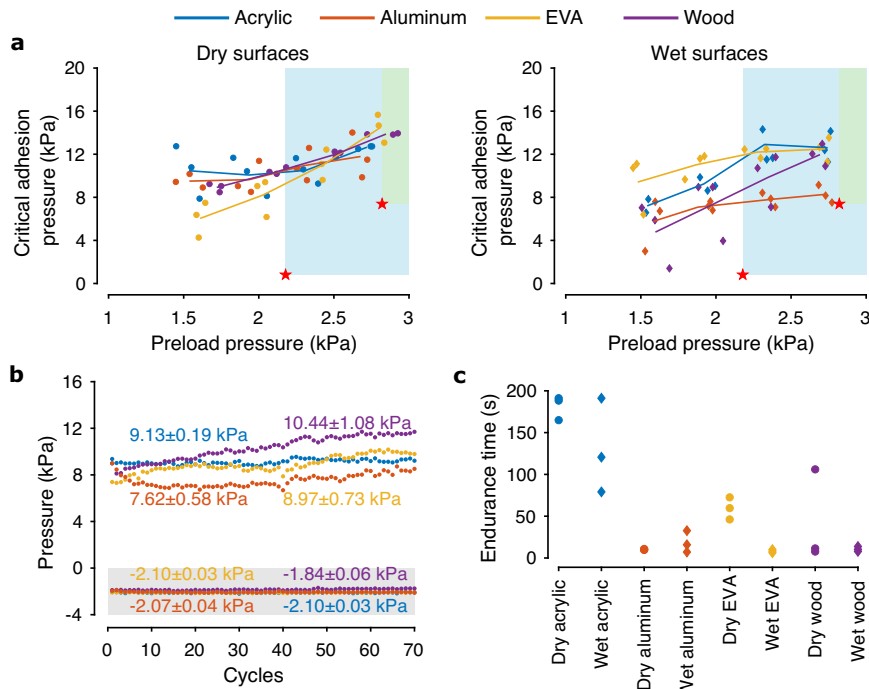

**Fig. 3 Experimental characterization of the adhesion.** Colors represent different substrates. Circle and diamond symbols refer to dry and wet substrates. **a** The critical adhesion pressures under different preload pressures. The lines show the average values from three measurements, illustrating the increasing trend. The red stars indicate the maximum preload pressure the robot can generate and the critical adhesion pressure for the ceiling (light blue shading) and wall (green shading) perchings (calculated according to the models). **b** Adhesive reusability test results. Negative pressure is preload and positive pressure is critical adhesion pressure. **c** Results from the endurance tests.

adhesion provides vertical support for the robot to stay on the wall. In such a stage, there exists a small gap between the ceiling adhesive pads and the surface (Fig. 2d). The joint rotation enables two types of adhesive pads to function independently in separate perching scenarios.

To evaluate the adhesion pressure required for the robot to remain perched on the wall, stress analysis is again considered. The analysis provided in Supplementary Note 2 shows that the average normal adhesion pressure required for the robot to stay attached to the wall is minimal as the weight of the robot is supported by the shear adhesion. Meanwhile, there exist lower and upper bounds for the maximum local adhesion pressures for the robot to be in moment equilibrium when perched and for the peel-off to be feasible. With the proposed mechanism, the tenfold difference between the lower and upper bounds simplifies the synthesis of the adhesive. Furthermore, for the adhesive with the same composition and maximum adhesion pressure to be universally compatible with ceiling and wall perchings, both Equations S3 (Supplementary Note 1) and S8 (Supplementary Note 2) must be simultaneously satisfied. In addition to the preload, distance parameters, the size of the wall and ceiling adhesive pads could be adjusted accordingly.

**Mussel-inspired wet adhesive.** The wet polymeric adhesive employed in this work displays several advantages over widely used dry adhesives and microspines. While gecko-inspired adhesives have demonstrated controllable and outstanding adhesion[56–58], they rapidly lose the stickiness on dampened surfaces. Microspines show reliability but require non-smooth surfaces for anchoring[34,58]. On the other hand, adhesion in the presence of water is effortless for marine mussels, owing to their proteinaceous secretions rich in 3,4-dihydroxyphenyl-L-alanine (DOPA)[52]. It was previously shown that DOPA-modified PSAs exhibited excellent adhesion both in dry and wet conditions[42–45].

For these reasons, the wet adhesive synthesized for dry and wet perchings is a polymeric material bearing two random monomeric units: dopamine methacrylamide (DMA) and methoxyethyl acrylate (MEA). Undergoing a simple radical polymerization, p(DMA-co-MEA) was obtained with high yielding and excellent adhesion[42,44,45]. The synthesis of p(DMA-co-MEA) is described in Methods. The adhesive pads were obtained after bonding a thin p(DMA-co-MEA) layer to a thin laset-cut sheet of PDMS.

To quantitatively evaluate the suitability of biomimetic wet adhesive for ceiling and wall perchings, characterization experiments were performed to verify three relevant properties of the adhesive on four materials: acrylic, aluminum, EVA foam, and wood. These substrates, representative of common man-made and natural materials, vary widely in terms of hydrophobicity and roughness as characterized by static contact angles (Supplementary Fig. 3, all materials except EVA foam are hydrophilic) and images from a scanning electron microscope (Supplementary Fig. 4, acrylic appears distinctly smooth at 1 μm resolution whereas EVA foam is highly and unevenly porous with the pore diameters on the order of 10–200 μm) as reported in Supplementary Note 3. The adhesive characterization tests include measurements of the critical adhesion pressures, reusability, and creep behavior. The reusability test was conducted to verify that the adhesive retains its adhesion after dozens of cycles of attachment and detachment. The test on creep resistance was designed to evaluate the performance or endurance of the adhesive under tensile load. These characterization procedures were conducted with four candidate surface materials, both under dry and wet conditions at room temperature. The testing is according to the protocols in Methods.

The results reveal that the maximum adhesion pressures have a positive correlation with the applied preloads as anticipated (Fig. 3a). The trend is observed in all four materials tested in both

dry and wet states. The critical pressures were found to be 3–5 times as high as the preloads, and the performance difference in dry and damp states is relatively modest. The critical adhesion pressure of EVA foam, which was found to be hydrophobic, rose slightly when wet. This is in contrast to dry adhesives, of which the stickiness is severely demoted on damp surfaces. The findings also verify that, in addition to the size, preload forces must be taken into account during perching.

In regard to the reusability, a single adhesive pad displayed no notable variation in the adhesion pressure over 70 cycles of preloads and detachments on each surface material (Fig. 3b). However, under prolonged tensile pressure, the adhesive may deform, leading to an undesired detachment when paired with aluminum or wood. The times the adhesive pads stayed attached to the surfaces are presented in Fig. 3c. The outcomes suggest the need for periodic reinforcement of compressive pressure to counter the deformation for the deployment of the adhesive in the ceiling perching task.

**Proximity effect**. In this work, when the robot perches on a wall or ceiling with the rotors' axes normal to the surface, the propellers are located within a few millimeters or less from the surface and the upstream wake is substantially interrupted. The corresponding proximity effect, often referred to as the *ceiling effect*[41,47,48], markedly affects the aerodynamic performance of the propellers. Previously, the ceiling effect has been taken into account to stabilize MAVs when they fly near a ceiling[59,60]. However, it has not been leveraged for wall and ceiling perching tasks as evidenced in refs. [31,32,34–39] (see also Supplementary Table 1).

As defined in Supplementary Note 4 and Supplementary Fig. 5, $\gamma(d) \geq 1$ is the proximity effect coefficient that describes the factor of reduction in the mechanical power required by a spinning rotor to generate the same magnitude of thrust when it is at distance $d$ from the surface. For the developed prototype ($d \approx 2$ mm), experimental measurements manifest the proximity coefficient of $\gamma = 2.72$. The result suggests a reduction in the overall power consumption of the robot for the thrust-assisted ceiling perching up to a factor of $\approx 2.2$ (calculated based on the analysis in Supplementary Note 4, the difference is attributed to the power loss in the motors and electronics according to the diagram in Supplementary Fig. 6). When combined with the support from the adhesive pads (for instance, the decrease in the power required to generate thrust $T$ to augment $2F_c$ in Equation (1)), the overall power consumption during perching can be substantially decreased.

**Ceiling perching**. The strategy for perching on an overhang makes use of the combined normal adhesion force and the propelling force, reinforced by the proximity effects, to counter the robot's weight. With the propellers remaining active at low thrust commands when the robot is perched, the peel-off is obtained by lowering or powering off the propellers to increase the local maximum tensile pressure[61]. With assistance from the propelling thrust, no additional mechanism is required for preloading or disengaging the adhesive pads. In addition to an easy and reliable detachment from the surface, this thrust-assisted method reduces the dependence on the precision of the adhesion pressure, with the power consumption while perching further reduced by the surface-induced aerodynamic interactions.

The four-stage ceiling perching framework (Fig. 4) enables the robot to autonomously and reversibly perch on an overhang and conserve energy over a wide range of substrates. Leveraging onboard feedback, the maximum adhesion pressure of each particular surface is evaluated on-the-fly to reduce the perching power. As a result, a human pilot only needs to (i) initiate the perching sequence when the robot hovers below the surface; and (ii) instruct the robot to take off afterward. During the process, the robot relies primarily on its IMU measurements for surface detection, command turning, and stabilization using the method described in Supplementary Note 1. Position feedback from the motion capture system is used for taking off, but this can be substituted by the onboard feedback as detailed in Supplementary Note 5.

The devised ceiling perching strategy was implemented for the robot to demonstrate the surface locomotion on four different materials, including acrylic, aluminum, EVA foam, and wood, in both dry and wet conditions (Fig. 5 and Supplementary Figs. 7–14 and Supplementary Movie 2). To illustrate the effectiveness of the method, the detailed results and associated data of the dry acrylic case are exemplified (Fig. 5a, b and Supplementary Fig. 7). In the first stage, the robot was commanded to hover below the overhang (Fig. 5a). The contact was made and detected through the onboard accelerometer (refer to Supplementary Fig. 15) shortly after the thrust command was elevated. The flight altitude was not directly controlled during the approach.

In Stage II, the robot briefly generated preload (visible as intensified motor voltages in Fig. 5b). This was immediately followed by the first step of the command tuning process. The collective thrust was gradually decreased while maintaining a fixed moment arm $d_{ct}$ to the default value (all motors supplied with the same voltage) until the peel off was detected by the gyroscope (an example of the monitored roll rate shown in Supplementary Fig. 15) and the critical perching torque $\tau_c^* = T d_{ct} = mg d_{cg}$ from Equation (2) was determined (Fig. 5b). Next, the robot re-applied the preload and proceeded to increase the moment arm $d_{ct}$ and lower $T$ by altering the distribution of motor commands (keeping the torque fixed). In this circumstance, the robot stayed adhered to the ceiling when $T$ was reduced and $d_{ct}$ was maximized. The detachment did not occur as the conditions for force equilibrium (Equation (1)) was never violated due to the large adhesion pressure and high creep resistance between the adhesive pads and the acrylic (Fig. 3a, c). The robot then entered and spent approximately 10 s in the third power-conserving stage. Thereafter, the robot took off (Stage III) from the overhang by peeling off the adhesive and resumed flight. This was reflected as a brief drop in motor voltages before bouncing back to the hovering level. The perching maneuver was repeated to showcase the reliability of the adhesive and the strategy.

For situations involving a more challenging surface condition (lower adhesion pressure and sizable creep), the adhesive pads were unable to support the weight of the robot for an extended duration after $d_{ct}$ was raised and $T$ was decreased during the second tuning phase. This, for instance, is the case for the ceiling perching on wet EVA form (see Supplementary Fig. 12), of which the adhesive test suggested relatively poor resistance to the time-dependent deformation Fig. 3c). In such cases, the robot undesirably detached from the surface after a couple seconds. This was addressed by reverting $d_{ct}$ back towards the default value (moving the center of thrust back towards the center of the robot) to reduce the load on the adhesive. This alleviated the problem and enabled the robot to stay perched on the surface without an unintended detachment afterward.

**Wall perching**. Perching on the wall leverages shear adhesion to support the robot weight and significantly reduce the collective thrust to achieve equilibrium of moments and power conservation. The developed strategy, elaborated in Fig. 6a and Supplementary Note 2, renders the wall perching operation reliable by allowing the robot to momentarily apply substantial compressive

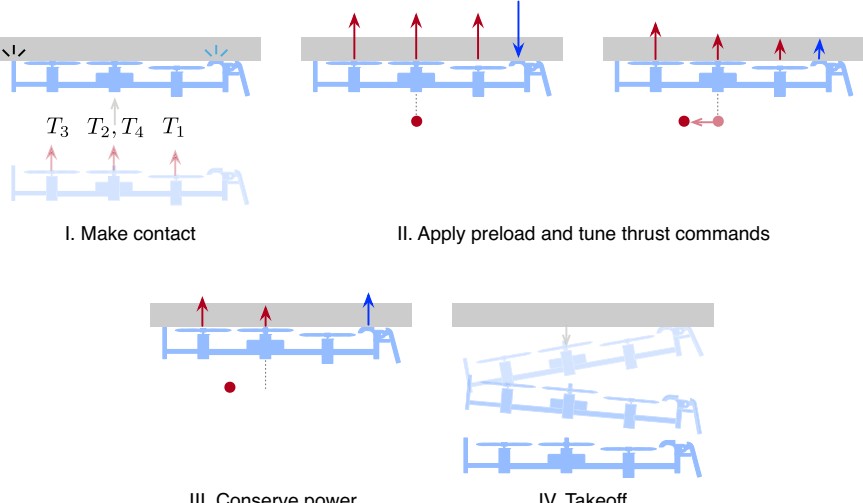

**Fig. 4 A four-stage ceiling perching strategy involving the manipulation of four rotor thrusts: $T_1$, $T_2$, $T_3$, and $T_4$.** Starting below a horizontal overhang, the robot first ascends to establish contact. Then, the adhesive is preloaded by momentarily applying maximum motor commands, with propelling forces further amplified by the proximity effect. Next, to conserve power, the robot adapts the thrust distribution, moving the center of collective thrust (depicted as red dots) away from the adhesive pads to lower the overall force commands. Lastly, takeoff is accomplished through peeling, requiring no additional mechanism or actuators.

preload to the adhesive, taking into account the actuation limit and the design and configuration of the robot (Supplementary Fig. 16). Once completely perched, the robot retains its stability with minimal propelling thrust, markedly lowering the power consumption compared to the regular hovering flight. Similar to ceiling perching, the developed wall perching method and control law allow the robot to perch autonomously. The motion capture system used for the takeoff can be replaced by onboard sensors as described in Supplementary Note 5.

With the proposed strategy, the robot demonstrated consecutive wall perching on a range of artificial surfaces: acrylic, aluminum, EVA foam, and wood, in both dry and wet conditions (Fig. 7, Supplementary Figs. 17–24, and Supplementary Movie 3). In the example of wet wood (Fig. 8a), the surface was dampened immediately before the attempts by a mist sprayer. After making an initial contact with the surface in Stage I, the roll angle was controlled to stay at $\theta$ near $\theta^* = 47°$ to apply preload. The horizontal thrust component was reinforced by decreasing $d_{jt}$ (Fig. 6b) by employing suitable thrust distribution, resulting the anticipated local compressive pressure of up to 2.8 kPa. In the third stage, the setpoint roll angle was gradually increased to 90°. The robot, in the perched state, only lightly actuated the far propeller ($T_3 \geq 0.05\ mg$) to remain in equilibrium (maximizing $d_{jt}$ Fig. 6b) and the adhesive pads were in light compression. The required thrust was significantly lower than that of for flight. The robot spent several seconds attached to the wall in the power conserving mode. To take off, the propellers were briefly stopped and the vehicle was controlled to roll down. When the roll angle was below $\alpha^-$ or 15°, the adhesive peeled off and the robot returned to flight before repeating the entire perching sequence again without landing (Fig. 8b). The same method was applied when the vehicle robustly perched on all dry and wet materials (Fig. 7 and Supplementary Figs. 17–24). Reliability and repeatability are attributed to the insensitivity to the required adhesion pressure of the method and the stability of the wet adhesive.

**Power conservation and flight endurance**. To assess the amount of power conserved by the hybrid perching method, benchmark hovering flights were performed on the original Crazyflie 2.1 robot with no hardware modification and the proposed robot equipped

with the custom-made perching mechanism. The weights of both robots, including markers for the motion capture cameras, were 31.06 g and 32.15 g. The proposed robot consumed marginally more power (8.6 W), on average, than the original Crazyflie 2.1 (8.3 W) owing to its heavier mass (see Fig. 9a and Methods for the measurement methods). The minute difference results in comparable flight endurance. The flight time was recorded as 455 s and 420 s for the proposed and original robots (Fig. 9b). The small difference in the flight times is likely due to other variations (such as the reliability of the batteries) rather than the difference in the flight power.

The power consumed by the robot when perching on ceilings and walls was calculated from flight data as detailed in Methods. The consolidated results (Fig. 9a) show that the robot expended between 4.7 and 6.4 W when it perched on an overhang and between 1.3 and 1.9 W when it perched on a wall, depending on the surface materials and conditions. Compared to flying, ceiling and wall perchings conserve approximately 40% and 80% of power, overall.

During ceiling perching, the degree of power conserved was dependent on the normal adhesion pressure. The power expended was further decreased by the proximity effect. Nevertheless, the overall power consumption includes not only aerodynamic and mechanical power, but also losses in the motors and electronics. To this end, the ceiling perching powers for dry and wet acrylic (4.7 W, Fig. 9a) were found to be the lowest among four tested materials, due to their relatively high normal adhesion (Fig. 3a) and creep resistance (Fig. 3c). In contrast, for the robot to stay perched on a ceiling padded with dampened EVA foam, it had to overcome the time-dependent deformation and primarily rely on the propelling thrust (see Supplementary Fig. 12). The proximity effect majorly contributed to the conservation of power (5.5 W, Fig. 9a), relegating the role of the adhesive pads to preventing slippage from shear adhesion. The situation highlights the advantage brought by the hybrid strategy, verifying that significant power conservation is still attained when the adhesive pads become ineffective.

Unlike the ceiling perching, the robot showed little variation in power consumption while perching on walls (Fig. 9a). This is explained by Equation (3), the weight of the robot during wall perching is entirely supported by the shear adhesion $F_w$.

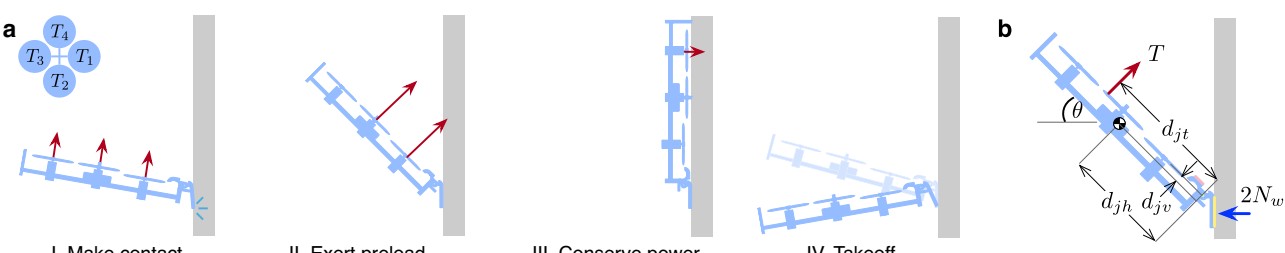

**Fig. 5 Ceiling perching experiments. a** Sequential images of the robot perching and taking off from the dry acrylic. **b** The time course of the motor voltages and the power consumption of the robot. Blue dots correspond to the timing of the images in **a**. Gray shadings indicate different perching stages. **c** The photographs of the ceiling perching experiments with different surfaces and conditions.

**Fig. 6 Wall perching maneuver and dynamics. a** A four-stage wall perching maneuver. **b** A free-body depicting the force (collective thrust $T$, weight $mg$, and surface normal $2N_w$) and moment (arm lengths $d_{jt}$, $d_{jv}$, and $d_{jh}$) contributing to the state of the robot during the perch-up maneuver.

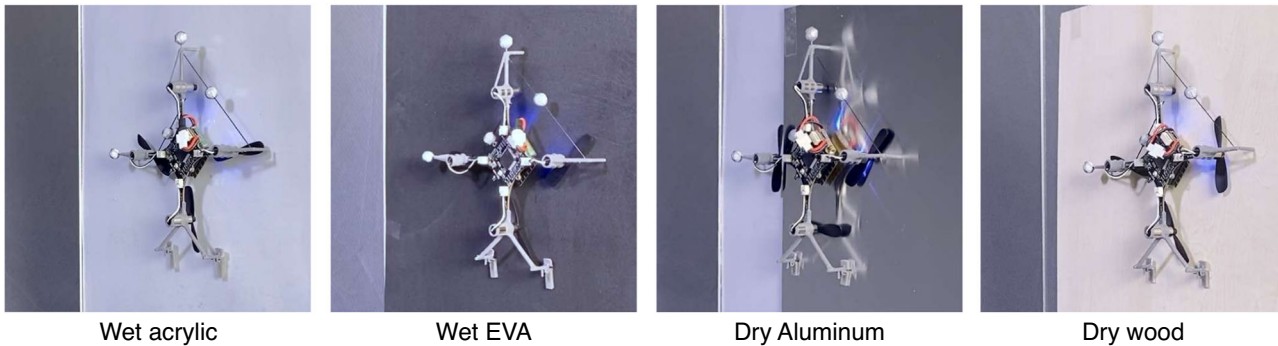

**Fig. 7 Example images of the robot taken from wall-perching flights.** The robot successfully perched on four materials and conserved power in both dry and damp conditions.

Meanwhile, the creep resistance of the biomimetic adhesive became less relevant as the adhesive remained under slight compression during the process. To this end, the robot was able to lower the power expenditure to around 1.3 W to 1.9 W when perching on the walls, with a notable portion (approximately 20–30%) of the energy dissipated by flight avionics.

Further, two prolonged endurance test flights were conducted as described in Methods (Supplementary Movie 4). In the ceiling perching trial, the robot perched and remained on dry aluminum shortly after taking off for over 790 s. The total flight time including landing was over 810 s (Fig. 9b). The average power of the robot during the perching period was 4.3 W (Fig. 9a). Similarly, for the wall perching test, the robot spent almost 1800 s perched on an acrylic surface, with the total flight time over 1860 s (Fig. 9b). On average the robot consumed 1.2 W while staying on the wall (Fig. 9a), which is one-seventh of the hovering power (8.6 W). In this circumstance, the avionics accounted for over 30% (0.38 W) of the dissipated energy. Compared to the hovering flight, the devised hybrid perching strategy dramatically extend the operational endurance. The mission times of the robot increased by 80% and 300% (see Fig. 9b), thanks to the combined adhesive forces and proximity effect.

## Discussion
In this study, we developed a strategy for a small multirotor vehicle to repeatedly adhere to and take off from multiple surfaces to conserve energy (Supplementary Movie 5). Unlike previous implementations of aerial-surface locomotion, the designed passive mechanism is lightweight and therefore does not adversely affect the overall aerial endurance while allowing the robot to perch on both horizontal and vertical wall surfaces. This has been accomplished by leveraging aerodynamic and adhesive forces. The direct use of propelling thrusts removes the need for sophisticated mechanisms or actuators for preloading and peeling off the adhesive. In the meantime, the actuation power is further and substantially reduced via the proximity effect. To this end, the prototype with the custom-made attachment mechanism is merely one gram heavier than the off-the-shelf robot. Yet, it demonstrated a fourfold increase in mission time when perching on a wall. The outcomes compare favorably against previous state-of-the-art solutions. The similar-sized aerial platform with microspines capable of rotor-assisted wall perching and climbing carried the additional 11-g mechanism, resulting in a substantially elevated flight power consumption [34]. On a larger scale, landing gears permitted robots to rest by grasping on a structure. However, the mass of the added component ($\approx$200 g in ref. [26] and $\approx$250 g in ref. [24]) inevitably elevated the flight power consumption. In addition, the reported adhesion pressure of $\approx$100 Pa is insufficient to be deployed with larger robots when taking into consideration the scaling of surface-to-mass ratio [39]. It can be

seen that the advantage of the proposed lightweight mechanism is its seamless integration into the commercial hardware that results in a minimal mass increase, unaffecting the regular flight performance.

In addition to leveraging the surface-propeller aerodynamic interaction, the use of mussel-inspired adhesive is also distinct from existing implementations. Unlike dry adhesives that require smooth and dry surfaces [56,58], the wet adhesive retains its effectiveness on damp substrates and sticks firmly to non-smooth EVA foam and wood (Fig. 3 and Supplementary Fig. 4; though, DOPA functionalized PSAs could be susceptible to oxidation under basic conditions and extreme temperature, i.e., below or far above the glass transition temperature [52]). This offers advantages over microspines that necessitate a relatively sophisticated mechanism and actuation for detachment from rough surfaces [27,34,58]. Other solutions, such as programmable polymer-based adhesive [55] or electrostatic adhesive [38,39], are limited to conductive or dry substrates, respectively. Through prudent design, the developed lightweight mechanism simplifies the ceiling and wall perching maneuvers. As a result, the approach can be readily deployed outside laboratory environments, only relying on onboard feedback as demonstrated in Supplementary Movie 6 (refer to Supplementary Note 5 for the implementation detail). Nevertheless, without a gripper or actuators [24,26,27], the limitation of the proposed solution remains is the reliance on structured horizontal and vertical surfaces. This strategy is, therefore, more suitable for urban usage. Therefore, the incorporation of the temperature-responsive moiety to attain temperature-controlled adhesion [53] could be considered to simplify the takeoff at the cost of increased weight and power. Another practical issue concerns the reusability or lifetime of the adhesive. Despite having displayed essentially no deterioration in adhesion pressure after 70 cycles of attachments and detachments, in real-world settings, some residual of the adhesive could occasionally be left on the surface, particularly when the interfacial molecular interaction between the adhesive and the substrate is high. For future improvement, these shortcomings can be alleviated through the adjustment of the chemical composition of the adhesive. It remains a challenging task to develop a single solution that universally excels in a large envelope of perching conditions.

## Methods
**Robot fabrication.** The robot with the designed attachment mechanism was fabricated using base parts of a Crazyflie 2.1 (a commercial micro quadcopter from Bitcraze). The components taken for the robot were the flight avionics (controller board), four $7 \times 16$-mm coreless DC motors, and propellers. The original plastic airframe parts were replaced by 3D printed components with supporting poles and placements for adhesive pads as shown in Fig. 1a (Gray resin, Form 3, Formlabs). Carbon fiber rods were used as the shafts of the revolute joints for the wall adhesive pads. Original 240 mAh Li-on batteries were substituted with 300 mAh batteries (applies to both the proposed robot and original Crazyflie 2.1 in flight tests). To realize indoor flights, four reflective markers were affixed on the robot with carbon

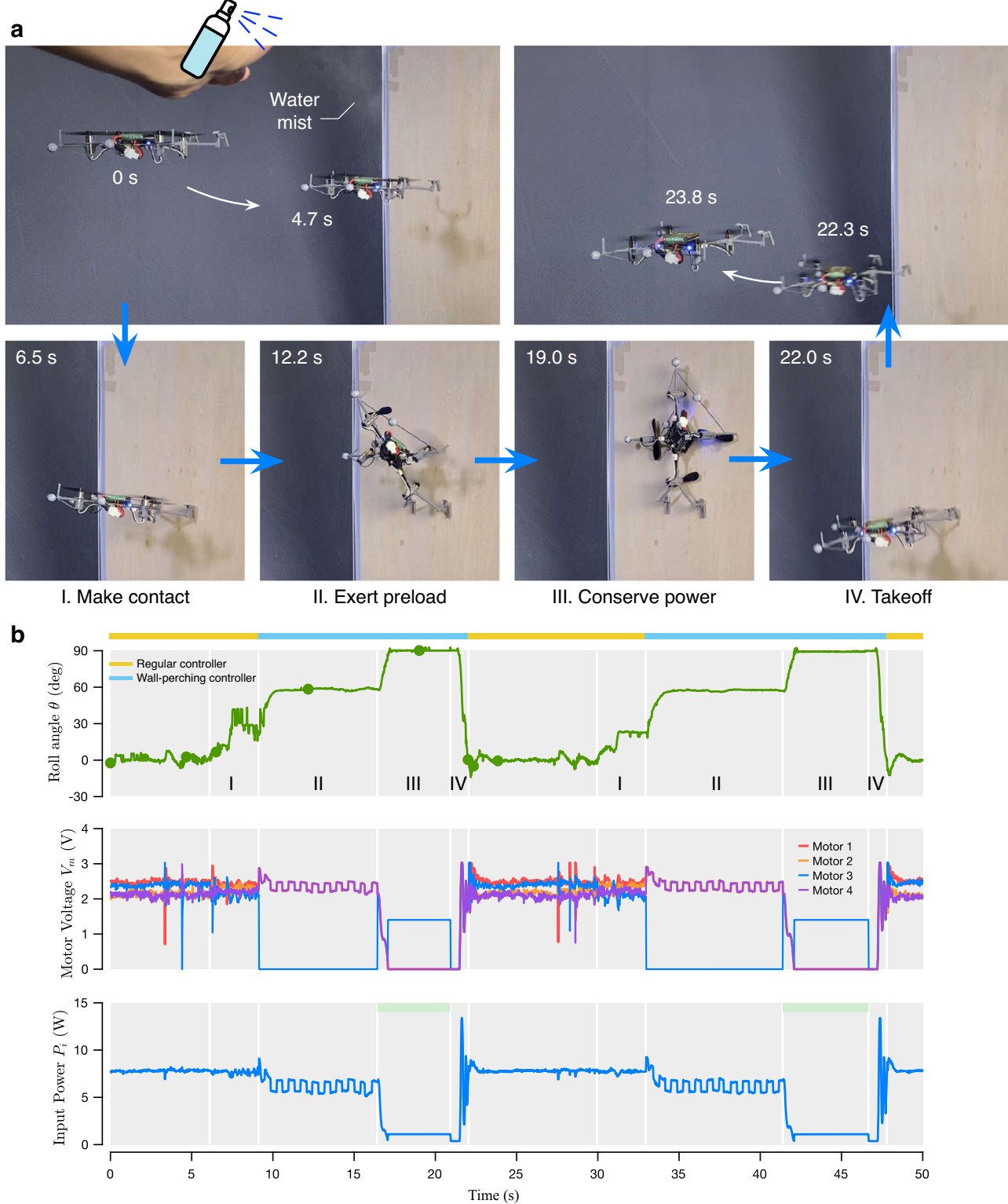

**Fig. 8 Wall perching experiments. a** Sequential images of the robot perching on a wood surface immediately after being dampened with a mist sprayer. **b** The plots of the perching angle of the robot, motor voltages, and power consumption during perching flight. The robot executed the wall perching and takeoff maneuvers twice in a single flight. Gray shadings indicate different perching stages. Green dots correspond to the timing of the image frames.

fiber beams. The parts were adhered using Cyanoacrylate adhesives and epoxy resin.

**Synthesis of wet polymer adhesive p(DMA-co-MEA) and fabrication of the adhesive pads**. The synthesis of the adhesive follows approximately the procedures described in refs. [42,44,45], but without nanopillar structures. Dopamine methacrylamide

(DMA) was synthesized according to the procedure in ref. [42]. First, 20 g of sodium borate and 8 g of NaHCO$_3$ were dissolved in 200 ml of deionized water and bubbled with N$_2$ for 20 min. Next, 10 g of dopamine HCl (52.8 mmol) was added, followed by the dropwise addition of 9.4 ml of methacrylate anhydride (58.1 mmol) in 50 ml of THF, during which the pH of solution was kept above 8 with addition of 1 M NaOH if necessary. The reaction mixture was stirred overnight at room temperature with N$_2$ protection. The aqueous mixture was washed twice with 100 ml of ethyl acetate and

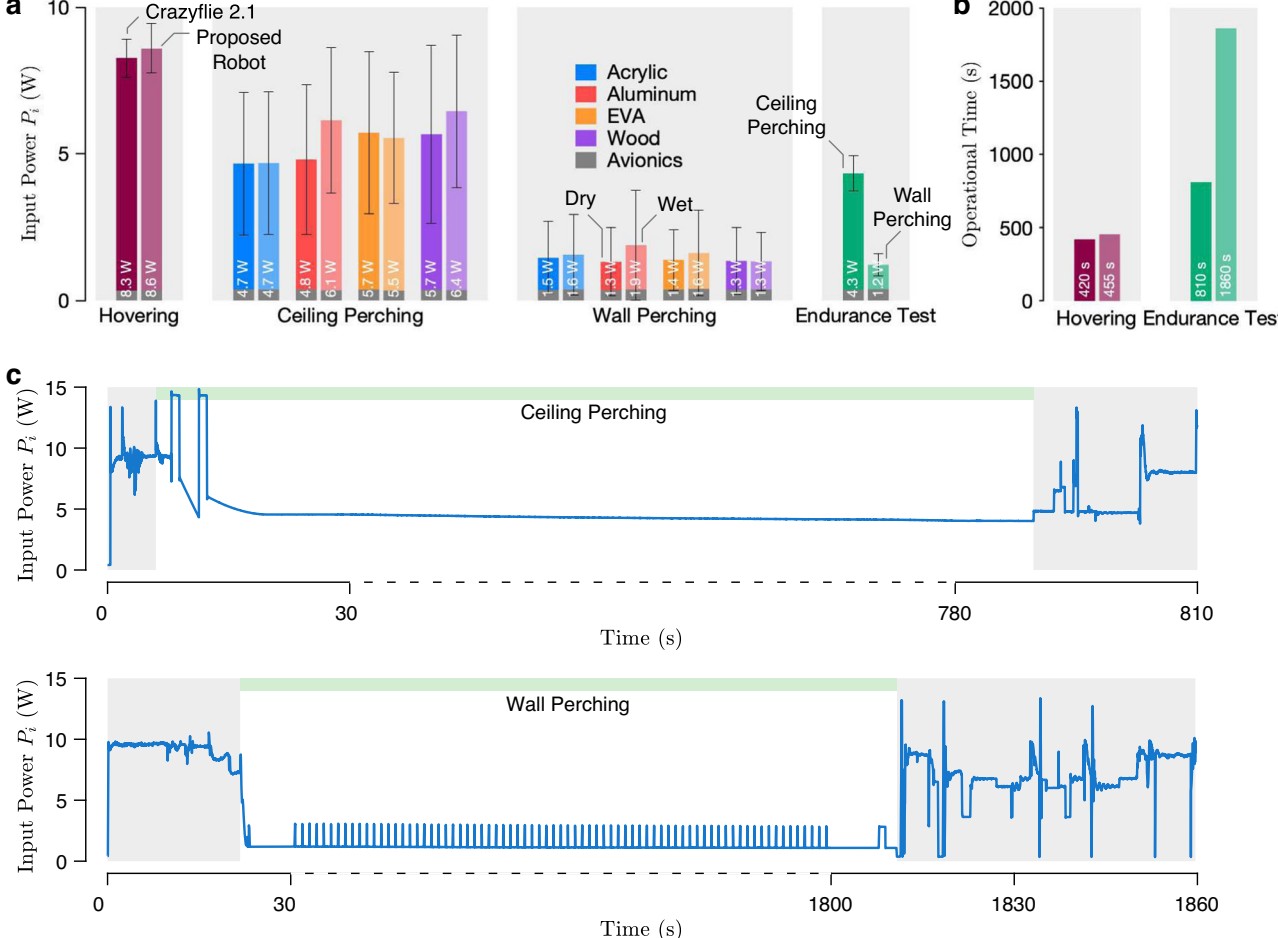

**Fig. 9 Power consumption and operational endurance. a** Average input power of the robots (i) from hovering flights, (ii) during ceiling perching, (iii) during wall perching, and (iv) during the ceiling and wall perchings in endurance flight tests, with error bars indicating one standard deviation (refer to respective experimental data for flight or segment durations used for the calculation). Gray shadings distinguish different sets of experiments. **b** Total operational times of the robots in extended hovering and perching flights (endurance test). **c** Plots of the power consumed by the robot during the endurance test flights. Gray shadings indicate different flight stages. The timescales in the middle portions, when the robot was perching, are sped up by factors of five and ten.

then the pH of the aqueous solution was reduced to less than 2 by concentrated HCl and extracted with 100 ml of ethyl acetate for three times. The final three ethyl acetate layers were combined and dried over $MgSO_4$ to reduce the volume to around 50 ml. 450 ml of hexane was then added with vigorous stirring and the suspension was held at 4 °C overnight. The product was recrystallized from hexane and dried to yield 6.4 g of light gray solid. p(DMA-co-MEA) was synthesized according to literature with slight modification[45]. Before polymerization, the inhibitor in MEA was removed by passing through a basic alumina column. 0.402 g DMA (1.8 mmol), 2.37 g MEA (18 mmol) and 40 mg of 2,2'-azobis(2-methylpropionitrile) (AIBN) were dissolved by 10 ml dimethylformamide (DMF) in a 50 ml three-neck round bottom flask. After $N_2$ bubbling for 30 min to exclude oxygen, the reaction mixture was heated to 60 °C and kept at this temperature for 3 h. The resulting viscous liquid was diluted with 10 ml methanol and precipitated by adding dropwise into 200 ml diethyl ether at 0 °C under continuous stirring. The resulting polymer was purified by redissolving in dichloromethane and reprecipitating in diethyl ether twice. The purified polymer was dried in a vacuum oven overnight at room temperature.

Small PDMS pads for depositing the synthesized polymer adhesive were fabricated from a thin PDMS sheet (thickness 1 mm), and cut to the specific sizes using a $CO_2$ laser cutter (Mini 24, Epilog). In this work, the use of flat PDMS bases produced sufficient adhesion pressure, dispensing the need for pillar structures to further boost the adhesion as evidenced in ref. [42]. To coat the adhesive on the laser-cut PDMS pads, the dried polymer was dissolved in chloroform at a concentration of 50–100 mg/ml. The solution was drop cast onto the clean and dry PDMS pads. After drying in the fume hood, the adhesive pads made of a thin p(DMA-co-MEA) layer tightly bonded to the underneath PDMS were obtained.

The complete adhesive structures were fabricated with 3D printed bases (Gray resin, Form 3, Formlabs) affixed to the synthesized adhesive pads through double-sided tape.

**Characterization of adhesion forces**. We constructed a test platform from a motorized linear stage and a 6-axis force/torque transducer (Nano 17 Titanium,

ATI) as shown in Supplementary Fig. 25a. The motion control and force/torque measurements were achieved through a computer running Simulink Real-Time (MATLAB & Simulink, MathWorks) with the data acquisition hardware (PCIe-6259, National Instrument). The platform was designed to be compatible with all of the tests described below with slight modifications.

**Critical adhesion measurements**. Uniform preloads were applied to the adhesive pads and the maximum tensile forces normal to the surface were recorded to obtain the critical adhesion pressure. To do so, the surface material was horizontally affixed above the transducer (resolution of 1/682 N or 0.15 gf). An adhesive pad (ceiling pad with the adhesive side pointing downward) was attached to the vertical motorized stage (see Supplementary Fig. 25a). A compliant shim (sponge) was inserted between the stage and the adhesive pad to facilitate the regulation of the normal force. During the test, the vertical stage steadily lowered the adhesive towards the fixed surface material (at 50 μm/s). Once in contact, the feedback from the transducer measured the compression as the preload force in real-time. The downward motion continued until the compressive force (preload) reached the preset value, at which point the stage stopped moving and held the position for 20 s. Thereafter, the stage reversed to an upward motion (at 500 μm/s) until the adhesive detached. The critical normal adhesion was registered as the maximum pulling force (refer to Supplementary Fig. 26).

We conducted the measurements of normal adhesion with four surface materials: acrylic, aluminum, EVA foam, and wood, in both dry and damp conditions. The preload pressures approximately at four preload forces from 61.7 to 125.5 mN, covering the anticipated preload pressure in both ceiling (2.2 kPa or 93 mN) and wall (2.8 kPa or 120 mN) perching (see Supplementary Notes 1 and 2). After every four measurements, the adhesive pad was replaced with a fresh sample. The process was repeated three times. Then, the same procedure was carried out for another surface material or another surface condition for a total of eight material/condition combinations, amounting to 96 datapoints in total.

**Reusability test**. The reusability test was carried out in a similar manner as the normal adhesion test, but the measurements were taken repeatedly without replacing the adhesive pads. Each fresh sample of a ceiling adhesive pad was tested consecutively 70 times, when subject to the preload pressure of 2.1 kPa or 90 mN.

**Creep resistance test**. The creep test was designed to characterize the impact of prolonged tensile force on the durability of the adhesive. We experimentally simulated the ceiling perching condition. During the ceiling perching, when $\xi_1$ is set to its minimum $\xi_1 = 0$ (most energetically efficient condition, see Supplementary Note 1), the pair of ceiling adhesive pads provide support of approximately 0.22 mg, equating to the adhesion pressure of 0.9 kPa or 3.5 gf for $A_c = 43$ mm$^2$.

The test setup is shown in Supplementary Fig. 25b. Different from earlier tests, the platform was inverted. The adhesive pad was attached on top of a 4.0 g counterweight (nearly equal to the required tensile force for ceiling perching) with the adhesive side up. The pad and counterweight were placed (but not fixed) on the vertical motorized stage, below the load cell. A sample surface material was attached to the bottom side of the sensor. When commanded, the stage lifted the counterweight and the pad toward the surface material to apply the preload of 9.5 gf (2.2 kPa, approximately the preload generated by the robot in ceiling perching, see Supplementary Note 1). The stage then displaced down, leaving the adhesive and the counterweight suspended from the surface material. This resulted in a constant tensile force on the adhesive. Over time, the adhesive deformed and eventually detached (see Supplementary Fig. 27 for an example). We took the time measurements for all four surface materials, in both wet and dry conditions. Three measurements were taken for each combination, amounting to 24 datapoints shown in Fig. 3b.

**Thrust and power measurements**. The measurements for power analysis were acquired through a benchtop setup (Supplementary Fig. 28a). For the experimental setup, the robot (Crazyflie 2.1, Bitcraze) was mounted on a transducer (Nano 17 Ti, ATI) for thrust force measurements. A removable acrylic plate was attached above the robot, acting as an overhang for generating the proximity effect when required. A data acquisition device (DAQ) (PCI-6229, National Instruments) for recording analog signals and a laptop running Crazyflie Python API were connected to a computer running Simulink Real-Time (Mathworks) as displayed in the schematics (Supplementary Fig. 29). The computer served as a central device for synchronization. The driving commands were sent from the computer to the laptop through UDP and then transmitted to the robot through the radio communication (CrazyRadio PA, Bitcraze). The robot's onboard firmware was modified to use the received commands to drive the motors directly. A current sensor (GHS 10-SME, LEM Inc.) was incorporated to measure the supplied current. The driving voltage was simultaneously logged via the onboard avionics and the DAQ.

First, to separately determine the power of flight avionics $P_{av}$, a power supply (GPD-3303S, GW Instek) was employed for the robot in place of the battery. The experiments were conducted by varying the supplied voltage from 2.6 to 4.3 V with a 0.1 V increment to simulate the varied battery voltage. Three tests were repeated at each voltage. A total of 54 tests were carried out to evaluate the consumed power while the robot was wirelessly communicating with the laptop. The robot was commanded not to drive the propellers in these tests. In each 40 s test sequence, the Crazyflie was switched on after 4 s and the radio communication started after 10 s. The communication then ceased at 30 s, before the robot was turned off at 33 s. The voltage and current were monitored throughout (example measurements from one test shown in Supplementary Fig. 28b). The voltage and power from each test were calculated by averaging the measurements taken between 17 and 27 s. The results from all 54 trials reveal that the flight avionics consumed an approximately constant amount of current (98.6 ± 4.8 mA), regardless of the driving voltage for the experimented range (Supplementary Fig. 28c). Therefore, the consumed power is approximately proportional to the supplied voltage (Supplementary Fig. 28c).

To determine the current, power consumption, and generated thrust of the propelling units, a high-current step-down power supply (QQYC-ZK-JVA-12KX, Q-BAIHE) was chosen for driving the robot. The measurements were carried out using the same setup (Supplementary Figs. 28a and 29). The thrust force was varied by altering the duty ratio of the pulse width modulation (PWM) signals the robot used to actuate the motors. A total of 111 PWM values, distributed between the duty ratio of 20% to 100%, which covers the operational range in flight, were experimented with. At each duty ratio, measurements of supplied current, voltage, and collective thrust were recorded (Supplementary Fig. 28b for an example with the duty ratio of 45%). The 10-s period from the 25-s sequence was used to calculate average onboard and DAQ voltages, current, and thrust. The force measurements from the first and last seconds, when the motors were not actuated, were used for evaluating the bias of the load cell. Voltages measured by the DAQ and the flight control board display a degree of disparity that can be reconciled via a simple model (see Supplementary Note 4). The entire process was repeated when an acrylic sheet was placed (but not in contact) over the robot as an overhang. The distance between the propellers and the sheet was tuned to emulate the actual distance in a perching flight. The measurements were taken to evaluate the impact of the proximity effect on thrust and power consumption.

For the calculation of power consumption, measurements from the DAQ were used when available. Otherwise, the model presented in Supplementary Note 4 and Supplementary Fig. 30 was applied to convert the onboard measurements to the DAQ-equivalent values.

**Hovering flight power**. The robots, with fully charged single-cell batteries, were commanded to hover until the onboard voltage dropped below 3.0 V. The flight logs (Supplementary Fig. 31), including motor voltages $V_m$ and onboard voltage $V_b$, were supplied to the developed model to compute the power consumption during the flights $P_i$ using the method described in Supplementary Note 4.

**Perching power**. For the ceiling perching, the portions of flight in both stages II and III were included (indicated by green horizontal bars in Supplementary Figs. 7–14 and Fig. 5c). For the wall perching, only the portion in stage III when the robot was up against the wall ($\theta = 90°$), after the application of preload, was considered (indicated by green horizontal bars in Fig. 8b and Supplementary Figs. 7–14).

**Endurance tests**. In both ceiling and wall perching endurance tests, the in-flight onboard voltages remained above 3.0 V at the end (Supplementary Fig. 32). The average power of the robot during the ceiling perching period was 4.3 W (taken from the region with a green horizontal bar in Fig. 9c), lower than the numbers obtained during short perching flights (Fig. 9a) due to the smaller proportion of the thrust-tuning stage. During the wall perching, the robot periodically reinforced the preload every 23.5 s by briefly stepping up the command of motor 1 to 2.7 V for 1.0 s to prevent the adhesive from peeling off as reflected in Fig. 9c. The average power was taken from the region with a green horizontal bar in Fig. 9c.

## Data availability

The authors declare that the data supporting the findings of this study are available within the paper and its supplementary information files.

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

## Acknowledgements

The work described in this paper was supported by grants from the Research Grants Council of the Hong Kong Special Administrative Region, China (project No. CityU 11205419) and the Shenzhen-Hong Kong-Macau Science & Technology Project (Category C) (contract number SGDX20220530111401009).

## Author contributions

Y.H.H., Y.C., Z.W., and P.C. conceived the ideas and designed the study. Y.H.H. and S.B. designed, fabricated, and tested the adhesive and robot prototypes. Y.Z. fabricated the adhesive. Y.H.H., S.B., H.J., and R.D. performed flight experiments. Y.H.H. and PC analyzed and model the data. Y.H.H., S.B., Y.Z., H.J., Y.C., Z.W., and P.C. wrote and revised the manuscripts.

## Competing interests

The authors declare no competing interests.
