## [Peer Review File · Communications Engineering]

Energy efficient perching and takeoff of a miniature rotorcraftReviewers' comments:

Reviewer #1 (Remarks to the Author):

This work presents a drone that can land on walls and ceiling structures. The authors propose utilizing adhesive pads fabricated from PDMS bases attached to a quadcopter's body to allow perching on structures.

I have a bipolar feeling about the overall story outlined in this article.

On the one hand, I am excited to see this work because perching on a structure is an interesting problem. It brings new capabilities such as prolonged reconnaissance from a perch (and many other merits). On the other hand, I am disappointed because it's hard to identify (and verify) the novelty of this work at the area the authors claim originality.

To be more specific, exploiting ground effects and attachment mechanisms of different types for perching in aerial vehicles has been reported before. This work is not presenting any new ideas compared to those. The overall robot hardware design and control concept introduced is not necessarily new either.

However, I believe the major novelty of the work could be using adhesive pads fabricated from PDMS bases. I do not have enough information in the paper to believe it outperforms existing solutions such as gecko- or mussel-inspired adhesives for dry and wet surfaces. For instance, the authors claim PDMS adhesive material used here performs well on wet and dry surfaces while they do not present any metrics of success based on which they make this conclusion.

What is considered a 'wet' or 'dry' surface? In other words, it is unclear what is the level of, e.g., surface porosity and roughness considered in these perch tests and how those are compared to other works. Simply categorizing these surfaces based on their materials (wood, aluminum, etc.) or spraying water to the surface in some random way, as it is presented in the videos, cannot be counted toward generating solid evidence to substantiate the claims made in this work.

Another criticism made here regarding capabilities when these PDMS adhesives are used in a perching drone is that the added value looks very marginal. For example, on a dry surface, existing gecko-style adhesive mechanisms can permanently keep an aerial vehicle attached to the surface, while the proposed solution requires some level of power consumption to keep the motor propellers running, as shown in the videos, which is very limiting. And, regarding perching on a wet surface, considering this vehicle's size, which is a small crazyflie quadcopter, simply mussel-inspired works can keep the vehicle on the perch permanently.

So, in terms of added capability, this idea does not have much to offer either. To conclude, if the major novelty of this work is the PDMS adhesive design, then why the narrative and title do not support it? Why is a small part of this draft on PDMS design? In my opinion, this article is not suitable for publication at its current status. The title has to be modified. The materials should be centered around the novel aspect introduced in the article. Besides, there are so much redundant data reported in the supplementary materials that could be easily removed without too much affecting the content of this draft.

Reviewer #2 (Remarks to the Author):

This paper introduces a method for perching and taking off walls and ceilings by means of an aerodynamic effect and a bioinspired adhesion pad. The method is validated with a 32-gram quadcopter. Experimental results show that the robot can repeatedly perform perching and takeoff

with the proposed method and achieve a power saving of 50% and 85% flight power when perching on ceiling and vertical wall, respectively.

The manuscript is clearly written and the novelty claims are backed by experimental data. There is limited statistical analysis of multiple flights (see also detailed comments on adhesion results), but the combination of modelling and experimental tests in my opinion are sufficient to back up the claims.

Major comments:

1. The authors' previous work [29] indicated that the ceiling could reduce the power consumption by a factor of 2 – 3 (Fig. 3a in ref. [29]). However, in this report the robot can save only 50% flight power with the help of the adhesive pads. It is therefore unclear to what extent the adhesive pads contribute to the power saving. How much power can be saved if only ceiling effect is taken into account in real flight tests?

2. It is not clear how the authors combined the ceiling effect and adhesion to optimize power saving. Adhesive pads contribute passive force to keep the drone attached to the ceiling, and this force should be maximized, but stay smaller than the mass of the drone to reduce propeller thrust and thus save power. I wonder if the authors considered this trade-off for the power optimization. Also, how did the authors select the surface area of the adhesive pad?

Minor comments and suggestions for improvement:

1) Page 3 Line 38 (Furthermore, both options must be deployed with a suitable mechanism for controllable adhesion.): You may want to rephrase the sentence. Adhesion is the interface attraction between two surfaces (<https://onlinelibrary.wiley.com/doi/full/10.1002/adma.201707035>). In the papers you mentioned, there is no systematic study presented on adhesion control. However, in the paper [25], researchers studied perching sufficiency. In paper [24], the spring-loaded mechanism was used to absorb an impact and store energy.

2) Page 9, lines 118-119: How adhesion augments the aerodynamic force?

3) Page 10 Line 133 (With the assisted thrust, there exists no minimum limit of the adhesion force required for the robot to stay perched on a ceiling.): What are the minimum time requirements needed to stick to the surfaces using ceiling and wall adhesive pads? In an article that you cited (<https://www.nature.com/articles/s41467-017-02387-2#MOESM1>, Supplementary 7), the adhesion strength of DOPA modified PSA varies a lot with the contact time, and it is 10 times smaller for a short contact than for contact with the duration of 120 seconds.

3) Page 11 Line 150 (When it comes to the wall perching, the incorporation of the passive joints serves two purposes: precluding the ceiling adhesive from sticking to the wall (and vice versa) and assisting the perch up motion towards the wall.): It may be difficult for a reader to imagine the impact of passive joints because the reader is not yet acquainted with the working principle of perching that you present later on pages 15-17. Is there a way to put the working principles of ceiling and wall perching (figures 4 and 6) before mechanical design?

4) Page 11, line 159: (To evaluate the adhesion pressure required for the robot to retain the perched up state. The stress analysis is again considered). English is awkward: please rewrite.

5) Page 11 Line 161 (The analysis provided in Note S5 shows that the normal adhesion pressure required for the robot to stay attached to the wall is minimal.): What do you precisely mean by "minimal"? From common sense, one may think that the normal adhesion force has to be higher or equal to the mass of the drone.

6) Page 12 Line 175 (It was previously shown that DOPA modified PSAs exhibited excellent adhesion both in dry and wet conditions): Are there any limitations of DOPA modified PSAs? As mentioned

earlier, the surface temperature can significantly change the adhesion properties. Also, there is a relationship between applied adhesive force (force you apply to stick the DOPA modified PSA onto the substrate) and adhesive time on the resulting adhesive force.

7) Page 12 Line 179 (The adhesive pads were obtained after bonding a thin p(DMA-co-MEA) layer to a thin laser-cut sheets of PDMS.): It has been shown that the casting of p(DMA-co-MEA) layer on a pillar structure resulted in a better adhesion rather than a flat sheet (<https://www.nature.com/articles/nature05968>). Why have you decided to use a simple layer design instead of using a more beneficial pillar design? Also, different material compositions of DOPA modified PSAs can have significantly different dry and wet adhesive properties with a specific substrate (<https://pubs.acs.org/doi/full/10.1021/acsami.9b08429>). In this work, the researchers compared different material compositions for dry and wet adhesion for steel and high-density polyethylene. In your work, you tested the adhesion with a wide range of substrate materials. How did you choose those materials?

8) Page 14 Line 192 (The trend is observed in all four materials tested (acrylic, aluminum, EVA foam, and wood) in both dry and wet states.): For DOPA modified PSAs (<https://www.nature.com/articles/s41467-017-02387-2#MOESM1>), as well as for every other PSA (<https://onlinelibrary.wiley.com/doi/full/10.1002/adem.202200355>), the temperature of the adhesion characterization testing protocol is crucially important as it significantly changes the adhesion force between the PSA and substrate material. I did not find any information about the temperature of the interface between PSA and a substrate as well as the temperature of the environment. Can you provide it and show how these temperatures influence the adhesion behavior?

9) Page 22, lines 291-301: The proposed mechanism is heavier than the original Crazyflie 2.1. I wonder why it displays longer flight time (455s) compared to flight time of the unmodified drone (420s).

10) Page 22, Figure 9C is missing.

11) Page 25 Line 375 (For future improvement, these shortcomings can be alleviated through the adjustment on the chemical composition of the adhesive.): DOPA modified PSAs have different adhesion properties at 25 and 40 °C (<https://www.nature.com/articles/s41467-017-02387-2#MOESM1>). The control of the surface temperature allows changing the adhesion by 3 times from 0.5 to 3kPa. Why have you decided to peel off the material instead of using its intrinsic adhesion temperature dependence? In future work, you may also try to build an adhesive pad with a controlled interfacial temperature. Such a type of pad can switch the adhesion force from low (at room temperature) to high (at 40 °C) requiring less energy for landing and taking off. This improvement can help you to spend even less energy during landing and taking off reducing the power consumption of the drone.

Reviewer #3 (Remarks to the Author):

Dear authors,

The research work proposed is interesting and the methodology seems sound. Also the work amount to perform the experiments and present the results is significant. However, I think this paper can be accepted only after minor revisions.

My remarks and suggestions will mainly concern the vehicle and flight physics.

- Some statements lack of numerical values for example line 215-216 "experimental measurements manifest the proximity coefficient of $\gamma = 2.72$, indicating a significant reduction in the power

consumption ", what is a significant reduction? 2? 10?

- In Figure 4, you mention "the adhesive is preloaded by applying maximum motor commands", then "To conserve power, the robot adapts the thrust distribution". To conserve the same power as "maximum motor commands"? The objective I guess is to use the minimum power to stay perched which is contradictory to what I read. More explanations and reformulation should be given.

- In Figure 4, the take off phase is not clear. In the text you mention "thrust-assisted method" and we do not see any thrust in the figure. Is it in the same direction (towards the ceiling) as for contact but lower thrust level? Or in the opposite direction (towards the ground) with low or high thrust? Again this part is missing numerical values. Figure 5 (B) does not answer these questions either. Adding an estimated thrust plot using a simple model with measurements as inputs could be useful.

- The control strategy or control laws are not presented. I wonder if the UAV is remotely piloted or autonomous? This is an important information to give and develop in the paper.

Summary of Major Changes

In this submission, we majorly revised the manuscripts in three aspects as suggested by the reviewers. To facilitate the review, we provide a brief summary of changes as follows.

First, we have substantially re-written the introduction to highlight the novelty of this work and clearly point out the drawbacks of previous perching methods for small aerial robots. In particular, we collate 22 papers into a table and plot out the mass ratio of the perching mechanisms and the perching mode to highlight that the proposed hybrid method is the only one that does not make the robot significantly heavier. In addition, very few robots have shown an ability to perch on a ceiling.

Second, we systematically characterize the chosen materials for hydrophobicity and roughness by means of contact angle and SEM images. The outcomes show that the four representative materials vary widely. This is added as supplements with the summary of the results provided in the main text.

Lastly, we address the question on temperature sensitivity of the adhesive. In short, the adhesive used in this work is not very sensitive to temperature around the room temperature as its composition is different from some cited work. In addition, we have made minor revisions throughout as requested by the reviewers.

Response to Comments

For clarity, the provided comments are in blue. Our revised texts, taken directly from the revised manuscript, are shown in red. If there is anything further we can do to clarify any of these points, please let us know.

Reviewer 1

- Overall Comment:* “This work presents a drone that can land on walls and ceiling structures. The authors propose utilizing adhesive pads fabricated from PDMS bases attached to a quadcopter’s body to allow perching on structures.

I have a bipolar feeling about the overall story outlined in this article.

On the one hand, I am excited to see this work because perching on a structure is an interesting problem. It brings new capabilities such as prolonged reconnaissance from a perch (and many other merits). On the other hand, I am disappointed because it’s hard to identify (and verify) the novelty of this work at the area the authors claim originality.”

— We thank the reviewer for the evaluation of our work and constructive comments. On the issue of the originality, we believe it is partly because we failed to highlight some principles clearly in the introduction. Please allow us to address your concerns on various

1228 points as listed below.

- 1229 • *Major Comment 1*: “To be more specific, exploiting ground effects and attachment mech-
1230 anisms of different types for perching in aerial vehicles has been reported before. This
1231 work is not presenting any new ideas compared to those. The overall robot hardware
1232 design and control concept introduced is not necessarily new either.”

1233 – We appreciate your concern on the novelty of the ground effects and attachment mech-
1234 anisms. Please allow us to point out, in a technical manner, how our implementation is
1235 unique from existing works. There are four major points.

1236 First, while we cannot disagree that *perching* is not a novel concept, our implementation
1237 is unique. In evidence of this, we have carried out an extensive and systematic literature
1238 review. We identified 22 papers (listed in Table S1, reproduced below). All of them
1239 requires an extra actuator to allow the robots to perch and take off (except one [26], which
1240 perches by resting/anchoring on a branch). We do not need an actuator or spring-loaded
1241 mechanisms (how this is accomplished is explain in the third point).

1242 As a consequence, the weight added by our perching mechanism is merely 1.1 g (3% of
1243 the total mass). This is by far the lowest (in absolute and relative terms) as illustrated in
1244 Fig. S9 (newly added in this revision, reproduced below). With negligible added weight,
1245 the *flight* endurance (not perching time) is unaffected. We firmly believe that this is a new
1246 concept compared to previous works we have seen.

1247 Second, in terms of the attachment mechanisms. Among 22 citations, they are majorly
1248 spines, with a few cases of electroadhesion and dry adhesives. The ceiling effect has
1249 never been used for perching (the idea might have been mentioned; ground effects are not
1250 applicable in this case). The ceiling effect has been taken into account for flight stability
1251 and control when robots operate near a ceiling [58, 59]. However, as far as we are aware,
1252 it has not been comprehensively shown to *conserve power* in actual perching flights as
1253 we demonstrate in this work. In fact, it is integral to our approach.

1254 Third, on the topic of the mechanism designed for attaching and taking off, our strategy
1255 is entirely different. This comes from the fact that we combine the adhesive with the
1256 proximity effect. Therefore, the use of propelling thrust is essential. The robot needs to
1257 expend small amount of power while perching. This may sound contradictory but the
1258 amount of power needed is small thanks to the proximity effect. This topic on the trade-
1259 off between the power spent while perching and the weight of the mechanism has never
1260 been addressed. Since the mechanism in this work is entirely passive, it is extremely
1261 light. The resultant weight of the robot remains unchanged from the original design
1262 (32.15 g vs 31.06 g). The flight results show virtually no difference in the hovering power

1263 consumption (8.6 vs 8.3 W, 3%). This stands out from previous perching mechanisms.

1264 Lastly, out of 22 existing works (Table S1), 15 of them are for perching on branches. Only
1265 seven robots can perch on a wall or ceiling, with only ONE capable of perching on both
1266 walls and ceiling [36]. This means it remains extremely difficult to perch on flat surfaces.
1267 Our robot is able to perch both on ceilings and walls. If the reviewer believes we miss
1268 any important references, please do not hesitate to point them out.

1269 For these reasons, we hope the reviewer sees the advantage (and drawback) of our method.
1270 We still firmly believe that our approach is unique and is worth the dissemination to the
1271 wider community. To ensure that potential readers get the message, we have majorly
1272 revised the manuscript to point out these novelties more clearly as elaborated below.

1273 1) We have re-written the introduction. We categorize previous perching robots according
1274 to the perching surfaces (branches, walls, ceilings) and attachment methods (spines, elec-
1275 troadhesion, adhesives). More importantly, we include Fig. S9 to highlight the weights of
1276 those mechanism to show how they compare to our work.

1277 “Aerial-surface locomotion, or perching, emerges as a promising avenue that allows aerial
1278 vehicles to maintain a high vantage point for a prolonged period with less power consump-
1279 tion [17]. Among existing small flying robots with the ability to perch (see Fig. S9 and
1280 Table S1), actuated grippers are the most common mechanisms that enable the robots
1281 to grab on to branches [18-30]. Relatively few vehicles are able to land and take off
1282 from walls [31-36] and ceilings [36-38]. To establish a firm contact with flat surfaces,
1283 electroadhesion, [37, 38], gecko-inspired dry adhesives [34] and small needles, or mi-
1284 crosppines [31-33, 36], have been employed. With relatively weak adhesion pressure (less
1285 than 1 kPa), the use of electrostatic forces results in disproportionately large adhesive
1286 pads and is still limited to robots under 20 g [37, 38]. For dry adhesives and spines
1287 [31-34, 36], they were deployed with a servo or motor and suitable mechanisms (such
1288 as preloaded springs) to ensure the robots can detach afterwards (with an exception for
1289 the robot in [26], which directly anchors on a branch when perching). These additional
1290 components account for an appreciable portion of the final vehicle mass. As illustrated in
1291 Fig. S9, the perching mechanisms (including added actuators) constitute over 15% of
1292 the total mass for vehicles under 100 g. This elevates the power required for the robots to
1293 stay aloft. For multirotor platforms, momentum theory predicts the scaling between the
1294 aerodynamic power of a spinning propeller and the thrust T as $P_a \sim T^{3/2}$ [39, 40], im-
1295 plying that a 20% increase in weight, for instance, nominally leads to a 30% rise in power
1296 consumption. In other words, the introduced capability to perch and conserve energy
1297 simultaneously compromises the flight endurance significantly.”

Table S1 Examples of perching aerial vehicles and their attachment mechanisms

robots from	flight platforms	added weight	total weight	weight ratio	perching surfaces	attachments	perching mechanisms	actuators	others
Graule et al. [37]	flapping-wing	13.4 mg	97.4 mg	0.14	ceiling	electroadhesion	(without power autonomy) ¹		
Gomez-Tamm et al. [18]	flapping-wing	-	450 g	-	branch	claws	SMA springs (two claws, four fingers each)		
Kovac et al. [31]	fixed-wing glider	4.6 g	6 g	0.77	wall	spines		motor	spines
Desbiens et al. [32]	fixed-wing	28 g	400 g	0.07	wall	spines		SMA	elastic linkage
Stewart et al. [23]	fixed-wing	170 g	850 g	0.20	branch	claw/gripper		servo	spines
Estrada et al. [57]	multirotors	-	100 g	-	ground	microspines		servo	(for manipulation)
Zhang et al. [24]	multirotors	10 g	40 g	0.25	branch	compliant grippers,		motor	
Broers and Armanini [25]	multirotors	45 g	294 g	0.15	branch	soft grippers		servo	rubber band
Kirchgeorg and Mintchev [26]	multirotors	40 ² g	400 g	0.10 ²	branch	spines			elastic ribbon
Thomas et al. [27]	multirotors	158 g	658 g	0.28	branch	claw		servo	
Roderick et al. [28]	multirotors	250 g	750 g	0.33	branch	claw/gripper		motors	spines, tendon
Doyle et al. [29]	multirotors	478 g	1011 g	0.47	branch	gripping feet		servo	
Hang et al. [30]	multirotors	440 g	1560 g	0.28	branch	gripper		servo	
Nguyen et al. [19]	multirotors	140 g	1766 g	0.08	branch	grapple		motor	(motorized winch)
Popek et al. [20]	multirotors	372 g	2300 g	0.16	branch	gripper		motor	(manipulator)
Melaren et al. [21]	multirotors	551 g	-	-	branch	robotic hand		servo	tendon, springs
Liu et al. [22]	multirotors	920 g	3800 g	0.24	branch	fingers		motor	(four motorized fingers)
Pope et al. [33]	multirotors	11 g	37 g	0.30	wall	microspines		servo	bow spring (wall climbing)
Kalantari et al. [34]	multirotors	-	550 g	-	wall	dry adhesive		servo	spines
Tsukagoshi et al. [35]	multirotors	160 g	1700 g	0.09	wall	suction cups		servo, pumps	
Park et al. [38]	multirotors	3.4 g	20.4 g	0.17	ceiling	electroadhesion			
Jiang [36]	multirotors	15 g	150 g	0.10	wall, ceiling	microspines		servo	spines
This work	multirotors	1.1 g	32.15 g	0.03	wall, ceiling	bio-inspired adhesive, proximity effect			

¹ Not included in Fig. S9 due to the lack of power autonomy.

² Each spine module weighs 5 g. The estimated weight of 40 g assumes the robot has eight spine modules.

Figure S9: Mass ratio of the perching mechanisms (refer to Table S1 for itemized data and sources). The plot displays the mass of the mechanisms against the total mass of the robots in logarithmic scales, categorized by the perching ability. The dashed lines represent the weight ratios of the mechanisms with respect to the total mass.

1298 2) We state explicitly how the use of the proximity effect and the thrust assistance ren-
1299 ders the work distinct from previous works. This paragraph also outlines the working
1300 principles of the proposed method.

1301 “This work tackles the major shortcoming of existing perching methods for Micro Aerial
1302 Vehicles (MAVs) to rest on both an overhang or a wall. The proposed strategy, which
1303 combines the airflow-surface interactions [40] with mussel-inspired wet adhesive [41–
1304 44], dispenses the need for additional actuators for engaging and disengaging the mecha-
1305 nisms. When incorporated in to a 31-gram rotorcraft, the final mass of the robot is only
1306 3% heavier than the original prototype (32.15 g, see Fig. S9).

1307 To facilitate repeatable perching maneuvers without extra actuators, the adhesive pads are
1308 incorporated onto a lightweight customized airframe with a passive mechanism and the
1309 use of the proximity effect is integral. The developed framework differs from previous
1310 implementations in two aspects. First, to preload the wet adhesive, the propelling thrust
1311 is used directly (instead of the use of elastic energy stored in a mechanism as seen in [15,
1312 18, 23–26, 28, 31–33, 36]). This is feasible as the robot takes advantage of the aerody-
1313 namic effect induced by a nearby surface. The proximity effect [40, 45–47], akin to the
1314 well-known ground effect [46, 48], amplifies the propelling thrust by over a factor of two.
1315 Second, when the robot is attached to the surface, it is supported by both the adhesion
1316 force and small propelling thrust to stay in force and moment equilibrium. Despite the
1317 need for small thrust while perching, the power consumption is immensely reduced as the
1318 aerodynamic efficiency is notably boosted by the proximity effect. When instructed, fur-
1319 ther lowering or removing the thrust commands allows the vehicle to seamlessly detach
1320 from the surface by peeling off the adhesive. Therefore, the thrust assistance replaces the
1321 need for extra actuation or a sophisticated mechanism, rendering it suitable for small ve-
1322 hicles with limited payload. The strategy is compatible with both walls and ceilings. The
1323 developed palm-sized quadcopter, as a result, is able to substantially extend its mission
1324 time without compromising on the flight endurance.”

1325 3) In the main text under Section Results: Proximity Effects, we re-state that the ceiling
1326 effect has been studied, but never been shown to conserve power in actual perching flights.

1327 “... The corresponding proximity effect, often referred to as the *ceiling effect* [40, 46],
1328 markedly affects the aerodynamic performance of the propellers. Previously, the ceiling
1329 effect has been taken into account to stabilize MAVs when they fly near a ceiling [58,
1330 59]. However, it has not been leveraged for wall and ceiling perching tasks as evidenced
1331 in [31–38] (see also Table S1).”

- 1332 • *Major Comment 2:* “However, I believe the major novelty of the work could be using ad-
1333 hesive pads fabricated from PDMS bases. I do not have enough information in the paper

1334 to believe it outperforms existing solutions such as gecko- or mussel-inspired adhesives
1335 for dry and wet surfaces. For instance, the authors claim PDMS adhesive material used
1336 here performs well on wet and dry surfaces while they do not present any metrics of suc-
1337 cess based on which they make this conclusion.”

1338
1339 – We thank the reviewer for the constructive comment. There are two issues here. First,
1340 you are right to point out that the mussel-inspired adhesive used in this work has not been
1341 used for perching MAVs as we summarize in Table S1.

1342 Theoretically, the proposed method can be demonstrated with gecko-inspired dry adhe-
1343 sive (used in [34]). The mussel-inspired adhesive provides an added benefit as it does not
1344 require the surface to be totally dry. We agree that it does not *outperform* the dry adhe-
1345 sive. However, it is not our intention to claim that the use of this mussel-inspired adhesive
1346 is the main contribution of the work. In this regard, we duly cite all the works related to
1347 the development to the adhesive in the introduction. In the fabrication part, we provide
1348 a summary for completeness and refer readers to previous works. We believe it is still
1349 important to provide some context on the wet adhesive as it is still not widely used as the
1350 dry adhesive. We never claim that the wet adhesive generally outperforms dry adhesives
1351 and even if it does, the development of the adhesive is not the contribution of the work.
1352 Please refer to an excerpt from the introduction below, we only objectively state that the
1353 dry adhesive deteriorates in wet conditions. This is well-known fact.

1354 “While the hybrid perching strategy is compatible with different types of adhesion meth-
1355 ods, the choice primarily depends on anticipated environments. Microspines need sur-
1356 faces with prominent rugosity. The mussel-inspired wet adhesive offers certain benefits.
1357 Unlike dry adhesive pads constructed with hair-like microstructures, of which van der
1358 Waals adhesion rapidly deteriorates when dampened [49] (the susceptibility to water also
1359 applies to electrostatic adhesion presented in [37, 38, 50]), ...”

1360 Second, on the comment that “the authors claim PDMS adhesive material used here per-
1361 forms well on wet and dry surfaces while they do not present any metrics of success based
1362 on which they make this conclusion”, we would like to point out that we systematically
1363 and extensively quantified the adhesion pressure of the adhesive on various surfaces as
1364 shown in Fig. 3A (reproduced below). This is also in addition to published results [41]–
1365 [44] that we have already cited. Having said that, it is true that the definitions of ‘dry’ and
1366 ‘wet’ surfaces in this manuscript are not clear. We address this issue in our response to
1367 your Major Comment 3 below.

- 1368 • *Major Comment 3*: “What is considered a ‘wet’ or ‘dry’ surface? In other words, it is
1369 unclear what is the level of, e.g., surface porosity and roughness considered in these perch

Figure 3: Experimental characterization of the adhesion. (A) The critical adhesion pressures under different preload pressures. (B) Adhesive reusability test results. Negative pressures are preloads and positive pressures are critical adhesion pressures. (C) Results from the endurance tests.

1370 tests and how those are compared to other works. Simply categorizing these surfaces
1371 based on their materials (wood, aluminum, etc.) or spraying water to the surface in some
1372 random way, as it is presented in the videos, cannot be counted toward generating solid
1373 evidence to substantiate the claims made in this work.”

1374 – We thank the reviewer for raising this point. From our interpretation, it is not ade-
1375 quately scientific to merely categorizing surfaces according to material types. It is a valid
1376 concern.

1377 – We have made a major revision to quantitatively differentiate materials based on two
1378 measurements. First, to characterize the wettability, we measure contact angles between
1379 water droplet and the surface to see if the surface is hydrophobic or hydrophilic. This
1380 property is important as they are anticipated to react different with the mussel-inspired
1381 adhesive when wet.

1382 Second, in terms of surface roughness, we took SEM images at five magnification levels
1383 (unfortunately, we are incapable of constructing a 3D image from a stereo pair to properly
1384 measure the surface rugosity). The results testify that four materials vary significantly in
1385 surface roughness, with acrylic being extremely smooth, wood is fibrous with the fiber
1386 width of 5 – 20 μm , and EVA foam is visibly porous and nonuniform with the feature
1387 size ranging from 10 to 200 μm .

1388 The results of these two tests are summarized in the main text under section **Mussel-**
1389 **Inspired Wet Adhesive**.

1390 “To quantitatively evaluate the suitability of biomimetic wet adhesive for ceiling and wall
1391 perchings, characterization experiments were performed to verify three relevant proper-
1392 ties of the adhesive on four materials: acrylic, aluminum, EVA foam, and wood. These
1393 substrates, representative of common man-made and natural materials, vary widely in
1394 terms of hydrophobicity and roughness as characterized by static contact angles (Fig. S4,
1395 all materials except EVA foam are hydrophilic) and images from a scanning electron mi-
1396 croscope (Fig. S5, acrylic appears distinctly smooth at 1 μm resolution whereas EVA
1397 foam is highly and unevenly porous with the pore diameters on the order of 10-200 μm)
1398 as reported in Note S4. ...”

1399 The detailed test protocols and results are given as supplements and reproduced as fol-
1400 lows.

Surface Materials

Wettability

To characterize the wettability of the surfaces employed for the perching experiments, we measured the static contact angles of the chosen materials using a drop shape analyzer (DSA100, KRÜSS GmbH). A surface is deemed hydrophobic or hydrophilic if its static water contact angle is over or less than 90° . Prior to the tests, the samples were cleaned with ethanol and DI water and dried in an oven at 80°C . In each trial, once a droplet settled on the surface, left and right contact angles were optically recorded.

We conducted multiple tests for four investigated materials: five tests for acrylic (droplet volume: $0.55 \pm 0.06 \text{ mm}^3$), three tests for aluminum (droplet volume: $3.0 \pm 0.1 \text{ mm}^3$), five tests for EVA foam (droplet volume: $0.51 \pm 0.06 \text{ mm}^3$), and five tests for wood (droplet volume: $0.60 \pm 0.09 \text{ mm}^3$). Example photos of the measurements are shown alongside the measured angles in Fig. S4

Figure S4 Measured contact angles between water droplet and surface materials. (left) Bar plots of the measured contact angles showing the average and standard deviation values. (right) Optical images of the static droplets on four tested materials.

Surface Morphology

Four substrates were analyzed by a scanning electron microscope (FEI Quanta FEG 250, FEG-SEM) at five magnification settings, with the resultant images from three representative magnification levels shown in Fig. S5. In terms of surface roughness, we found the feature size of acrylic to be lower than $1 \mu\text{m}$. The images of aluminum show microridge-like structures that are approximately 1-5 μm apart. The images of wood suggest fibrous

1420
1421
1422

texture with the feature size of $5 - 20 \mu\text{m}$. The features are visibly less uniform. The images of EVA foam show the highest roughness, exhibiting micropores with highly inconsistent diameters of around $10-200 \mu\text{m}$.

Figure S5: Images from a scanning electron microscope.

1423
1424
1425
1426
1427
1428
1429
1430
1431
1432
1433
1434
1435
1436

– Regarding the remark “how those are compared to other works”, we have further emphasized the limitations of commonly used dry adhesive and microspines at places in the **Introduction**. The dry adhesive is limited to smooth surfaces and microspines is to uneven surfaces. Please note that, in those works, in which perching are shown, none of them quantitatively measured the surface roughness. Hence, with SEM images provided in this revision, we have already gone beyond a simple classification based on materials type as found in [31–34, 36].

“... For dry pressure-sensitive adhesives (PSAs) and spines [31–34, 36], they are limited to smooth (glass [34]) and rough surfaces (wood [31] and concrete [32, 33]), respectively...”

“While the hybrid perching strategy is compatible with different types of adhesion methods, the choice primarily depends on anticipated environments. Microspines need surfaces with prominent rugosity. The mussel-inspired wet adhesive offers certain advantageous. Unlike dry adhesive pads constructed with hair-like microstructures, of which van

1437 der Waals adhesion rapidly deteriorates when dampened [49] (the susceptibility to wa-
1438 ter also applies to electrostatic adhesion presented in [37, 38, 50]), the DOPA-based
1439 polymeric adhesive benefits from the wide range of workable surfaces (both smooth and
1440 rough, unlike commonly used dry adhesive and spines that are restricted to polished and
1441 uneven substrates, respectively) and the feasibility to be used passively. ”

1442 – Lastly, we totally appreciate the concern “spraying water to the surface in some random
1443 way, as it is presented in the videos, cannot be counted toward generating solid evidence to
1444 substantiate the claims made in this work”. In our defense, there is no practical, universal,
1445 and systematic method for quantifying the wetness of materials. There are methods for
1446 measuring the moisture content of wood or concrete using a moisture meter but that is not
1447 applicable to water content on surface of acrylic or aluminum. One could theoretically
1448 measure the amount of water (mass) per surface area but that does not ensure the uniform
1449 distribution or sizes of the droplets (particularly when the surface is vertical). The sprayed
1450 water may not all deposit on the surface, some may evaporate or be blown away by
1451 the rotor wake. There are numerous experimental factors that make the measurement
1452 challenging.

1453 But that is exactly why we did NOT merely present the flight experiments and the videos
1454 to make the claim. We systematically characterized the adhesive in a more controlled
1455 setting. For instance, the results in Fig. 3 evidently illustrate the change in adhesion
1456 pressure in wet and dry conditions. And the changes are dependent on the material types
1457 (and hydrophobicity). The adhesion pressure increases when the hydrophobic EVA foam
1458 is wet, in contrast to the other three hydrophilic surfaces. However, the adhesion pressure
1459 remain larger than the perching requirements in all cases.

1460 We truly hope that with the surrogate measurements of the adhesion pressure, reusability,
1461 and endurance, when combined with the tests of contact angles and SEM images, would
1462 be sufficiently compelling for the claim that the robot was able to perch on a wide range
1463 of surfaces and conditions. We again thank the reviewer for asking a critical question.

- 1464 • *Major Comment 4*: “Another criticism made here regarding capabilities when these PDMS
1465 adhesives are used in a perching drone is that the added value looks very marginal. For
1466 example, on a dry surface, existing gecko-style adhesive mechanisms can permanently
1467 keep an aerial vehicle attached to the surface, while the proposed solution requires some
1468 level of power consumption to keep the motor propellers running, as shown in the videos,
1469 which is very limiting. And, regarding perching on a wet surface, considering this vehi-
1470 cle’s size, which is a small crazyflie quadcopter, simply mussel-inspired works can keep
1471 the vehicle on the perch permanently.”

1472 – This remark is related to Major Comment 1 and 3 (two separate issues). First, we agree

1473 that the added value of the wet adhesive over dry adhesive may be not very significant,
1474 but it is still beneficial. This, however, should NOT be seen as the drawback as the main
1475 contribution of the work is not the wet adhesive (and we do not claim that as elaborated
1476 in the introduction).

1477 On the second issue, where the robot needs to consume some power while perching. This
1478 maybe considered the drawback of the work, but this simultaneously brings a significant
1479 advantage and it is where the method is unique from previous 22 papers listed in Table S1.
1480 It is because of this trade-off, the added weight of the perching mechanism is only 1.1 g
1481 or 3% of the final mass (this is significantly lower than the minimum of 15% for flying
1482 robots under 100 g shown in Fig. S9). Hence, the already short flight time of our robot is
1483 not compromised. The use of thrust, when taking advantage of the ceiling effect, allows
1484 us to remove a motor or servo to preload and peel of the adhesive. While the drawback
1485 can be considered limiting, it still extends the operation time of the robot significantly
1486 (2-4 times). Compared to the original robot, it is only beneficial. We have explained this
1487 in detail with our response to your Major Comment 1.

- 1488 • *Major Comment 5*: “So, in terms of added capability, this idea does not have much to
1489 offer either. To conclude, if the major novelty of this work is the PDMS adhesive design,
1490 then why the narrative and title do not support it? Why is a small part of this draft on
1491 PDMS design? In my opinion, this article is not suitable for publication at its current
1492 status. The title has to be modified. The materials should be centered around the novel
1493 aspect introduced in the article. Besides, there are so much redundant data reported in
1494 the supplementary materials that could be easily removed without too much affecting the
1495 content of this draft.”

1496 – We truly hope that at this point, the reviewer agrees with us that the contribution of the
1497 work is the *hybrid* perching strategy, where the combination of adhesive and the prox-
1498 imity effect extends the *operational time* of the robot without compromising on the *flight*
1499 *time*. This is the novelty and it is evidence-based as seen in Fig. S9 and Table S1. We do
1500 not highlight the features or the fabrication of the wet adhesive as the contribution of our
1501 work.

1502 –We have gladly taken your suggestion to trim down the main text and Materials and
1503 Method as appropriate, particular on the adhesive tests. Note that the materials and data
1504 reported in the supplements are the evidence and methodology on how we make the hy-
1505 brid approach feasible. We do agree that they are exhaustive (but not overlapping with
1506 the main text) and therefore are placed as supplements. This is also to comply with
1507 the journal’s policy on data availability. If you still believe any part of them is redun-
1508 dant, please point us out to specific sections and figures. We will be more than happy

1509 to shorten/remove them. We try to be comprehensive to make the work re-producible by
1510 other scientists/engineers. We appreciate your understanding.

1511 **Reviewer 2**

1512 • *Overall Comment:* “This paper introduces a method for perching and taking off walls
1513 and ceilings by means of an aerodynamic effect and a bioinspired adhesion pad. The
1514 method is validated with a 32-gram quadcopter. Experimental results show that the robot
1515 can repeatedly perform perching and takeoff with the proposed method and achieve a
1516 power saving of 50% and 85% flight power when perching on ceiling and vertical wall,
1517 respectively.

1518 The manuscript is clearly written and the novelty claims are backed by experimental data.
1519 There is limited statistical analysis of multiple flights (see also detailed comments on
1520 adhesion results), but the combination of modelling and experimental tests in my opinion
1521 are sufficient to back up the claims. ”

1522 – We appreciate the reviewer’s effort. We are glad to learn that you find our work is
1523 sufficiently sound. We try our best to address your concerns, please do not hesitate to let
1524 us know if you have further suggestions.

1525 • *Major Comment 1:* “The authors’ previous work [40] indicated that the ceiling could
1526 reduce the power consumption by a factor of 2 – 3 (Fig. 3a in ref. [40]). However, in this
1527 report the robot can save only 50% flight power with the help of the adhesive pads. It is
1528 therefore unclear to what extent the adhesive pads contribute to the power saving. How
1529 much power can be saved if only ceiling effect is taken into account in real flight tests?”

1530 – Great observation! The ceiling effect does reduce the *aerodynamic* and *mechanical*
1531 power consumption by a factor of 2-3 as characterized by the ceiling effect coefficient γ .
1532 Nevertheless, the saving in the overall power consumption is less as there are losses in
1533 the motors and electronics. This is why, in the end, only $\approx 50\%$ of the flight power is
1534 saved. We failed to point this out clearly. We have added the following sentences to the
1535 manuscript under Section **Proximity Effects**.

1536 “For the developed prototype ($d \approx 2$ mm), experimental measurements manifest the
1537 proximity coefficient of $\gamma = 2.72$. **Since this indicates the decrease in the mechanical**
1538 **power by the same factor, the result suggests a reduction in the overall power consumption**
1539 **of the robot for the thrust-assisted ceiling perching up to a factor of ≈ 2.2 (calculated based**
1540 **on the analysis in Note S3), the difference is attributed to the power loss in the motors**
1541 **and electronics). When combined with the support from the adhesive pads, the overall**
1542 **power consumption during perching can be substantially decreased.”**

1543 Since this is important. We re-state this fact when the results of the power consumption
1544 are presented under Section **Power Conservation and Flight Endurance**.

1545 “During ceiling perching, the degree of power conserved was dependent on the normal
1546 adhesion pressure. The power expended was further decreased by the proximity effect.
1547 Nevertheless, the overall power consumption includes not only aerodynamic and mechanical
1548 power, but also losses in the motors and electronics....”

- 1549 • *Major Comment 2*: “It is not clear how the authors combined the ceiling effect and adhe-
1550 sion to optimize power saving. Adhesive pads contribute passive force to keep the drone
1551 attached to the ceiling, and this force should be maximized, but stay smaller than the
1552 mass of the drone to reduce propeller thrust and thus save power. I wonder if the authors
1553 considered this trade-off for the power optimization. Also, how did the authors select the
1554 surface area of the adhesive pad?”

– We thank the reviewer for these important questions. What you said on the adhesion
force ($2F_c$) is almost spot on. To see how the adhesion and ceiling effect are combined,
this is captured by Eq. (1) in the main text (which states that “the ceiling adhesive pads
lower the required collective thrust.”):

$$2F_c + \sum_{i=1}^{i=4} T_i = 2F_c + T \geq mg, \quad (1)$$

1555 Based on this equation, the presence of $2F_c$ lowers the required thrust T . This immedi-
1556 ately lowers the power consumption. The ceiling effects allow the same thrust T to be
1557 generated with lower mechanical (and electrical) power. Hence, both contribute to the
1558 power conservation.

1559 To make this clear early in the manuscript, we have revised the following paragraph in
1560 the introduction.

1561 “This work tackles the major shortcoming of existing perching methods for Micro Aerial
1562 Vehicles (MAVs) to rest on both an overhang or a wall. The proposed strategy, which
1563 combines the airflow-surface interactions [40] with mussel-inspired wet adhesive [41]-
1564 [44], dispenses the need for additional actuators for engaging and disengaging the mecha-
1565 nisms. When incorporated in to a 31-gram rotorcraft, the final mass of the robot is only
1566 3% heavier than the original prototype (32.15 g, see Fig. S9).

1567 To facilitate repeatable perching maneuvers without extra actuators, the adhesive pads are
1568 incorporated onto a lightweight customized airframe with a passive mechanism and the
1569 use of the proximity effect is integral. The developed framework differs from previous
1570 implementations in two aspects. First, to preload the wet adhesive, the propelling thrust

1571 is used directly (instead of the use of elastic energy stored in a mechanism as seen in [15,
1572 18, 23-26, 28, 31-33, 36]). This is feasible as the robot takes advantage of the aerody-
1573 namic effect induced by a nearby surface. The proximity effect [40, 45-47], akin to the
1574 well-known ground effect [46, 48], amplifies the propelling thrust by over a factor of two.
1575 Second, when the robot is attached to the surface, it is supported by both the adhesion
1576 force and small propelling thrust to stay in force and moment equilibrium. Despite the
1577 need for small thrust while perching, the power consumption is immensely reduced as the
1578 aerodynamic efficiency is notably boosted by the proximity effect. When instructed, fur-
1579 ther lowering or removing the thrust commands allows the vehicle to seamlessly detach
1580 from the surface by peeling off the adhesive. Therefore, the thrust assistance replaces the
1581 need for extra actuation or a sophisticated mechanism, rendering it suitable for small ve-
1582 hicles with limited payload. The strategy is compatible with both walls and ceilings. The
1583 developed palm-sized quadcopter, as a result, is able to substantially extend its mission
1584 time without compromising on the flight endurance.”

1585 And since this is a critical point, we re-state this again in Section **Proximity Effect**:

1586 “... proximity effect ... When combined with the support from the adhesive pads (for in-
1587 stance, the decrease in the power required to generate thrust T to augment $2F_c$ in Eq. [1]),
1588 the overall power consumption during perching can be substantially decreased.”

1589 – In terms of “optimization”, ideally, larger adhesion is desired. However, in practice,
1590 this is not straight forward as the actual adhesion is highly dependent on the surface
1591 materials, wetness, preload, etc. Meanwhile, there are other important considerations
1592 when the peel off is considered so that the adhesion force need NOT be smaller than
1593 the mass of the drone. In the proposed design, the adhesive pads are displaced from
1594 the center of mass. Due to the moment arm (characterized by the mechanical advantage
1595 $d_{cg}/l_c \approx 10$), the robot is able to takeoff as long as the adhesion force is ≈ 60 times
1596 smaller than the weight of the robot (i.e. not just one time). This is shown by Eq. [S17]
1597 and the analysis in Note [S5.1]. This means the proposed method and design works for
1598 a wide range of adhesion force (from 0 to $60mg$). This makes selecting the size of the
1599 adhesive pads A_c in Eq. [S17] less difficult. In addition, since the actual adhesion pressure
1600 is dependent on the surface material and preload, the wide range of compatible adhesion
1601 pressure makes the method very reliable.

1602 To point this out in the manuscript, we added the info to Section **Mechanical Design and**
1603 **Adhesion Requirements for Detachable Perching** in the main text.

1604 “As detailed in Note [S5], the strategic placement of the ceiling adhesive pads amplifies
1605 the maximum tensile pressure the adhesive pads undergo in the peeling-off process by
1606 approximately 60 times compared to the case where the pads were put directly on top of

1607 the center of mass. As a consequence, the takeoff condition is relatively insensitive to the
1608 surface properties, applied preload, and the size of the adhesive pads. ”

- 1609 • *Minor Comment:* “Page 3 Line 38 (Furthermore, both options must be deployed with a
1610 suitable mechanism for controllable adhesion.): You may want to rephrase the sentence.
1611 Adhesion is the interface attraction between two surfaces ([https://onlinelibrary.
1612 wiley.com/doi/full/10.1002/adma.201707035](https://onlinelibrary.wiley.com/doi/full/10.1002/adma.201707035)). In the papers you men-
1613 tioned, there is no systematic study presented on adhesion control. However, in the paper
1614 [26], researchers studied perching sufficiency. In paper [23], the spring-loaded mecha-
1615 nism was used to absorb an impact and store energy. ”

1616 – We sincerely thank the reviewer for the suggestion. We agree that the phrase “con-
1617 trollable adhesion” may not be technically sound in this context because we meant to
1618 indicate the ability to perch and takeoff from surfaces (not altering the adhesive force).
1619 We decided to re-write this accordingly. Note that in while in [26], there exist elastic
1620 ribbons in the mechanism, they do not play a direct role for preloading or peeling off the
1621 microspines. The robot in [26] is unique as it is able to perch and take off directly through
1622 its weight and thrust (as it rests on branches, not walls or overhangs).

1623 In this re-submission, we have majorly revised the introduction to improve the manuscript
1624 as suggested by Reviewer 1. The problematic sentence has been replaced by the sentences
1625 below. If you have a suggestion on how we can better handle this, please do not hesitate
1626 to let us know.

1627 “... For dry adhesives and spines [31-34, 36], they were deployed with a servo or motor
1628 and suitable mechanisms (such as preloaded springs) to ensure the robots can detach
1629 afterwards (with an exception for the robot in [26], which directly anchors on a branch
1630 when perching)...”

- 1631 • *Minor Comment:* “Page 9, lines 118-119: How adhesion augments the aerodynamic
1632 force?”

1633 – Thank you for the question. Again, this important point is related to your second Major
1634 Comment. In simple terms, the adhesion force reduces the thrust required (compared to
1635 flying) for the robot to stay elevated (perched). The proximity effect reduces the power
1636 required to generate that thrust. They assist each other.

1637 To further clarify this, we edited the pre-cursory text. In the paragraph before the quoted
1638 statement, at the end of the previous subsection, we spell out the benefit of the proximity
1639 effect.

1640 “In both scenarios, as the propellers are close to surfaces, the proximity effect increases
1641 the aerodynamic efficiency of the propellers. When the input power or motor command

1642 is maintained, this leads to an increase in the propelling thrust. Alternatively, the intro-
1643 duction of a nearby surface lowers the power consumption of a spinning propeller when
1644 the thrust force is kept constant.”

- 1645 • *Minor Comment:* Page 10 Line 133 (With the assisted thrust, there exists no minimum
1646 limit of the adhesion force required for the robot to stay perched on a ceiling.): What are
1647 the minimum time requirements needed to stick to the surfaces using ceiling and wall ad-
1648 hesive pads? In an article that you cited ([https://www.nature.com/articles/
1649 s41467-017-02387-2#MOESM1](https://www.nature.com/articles/s41467-017-02387-2#MOESM1), Supplementary 7), the adhesion strength of DOPA
1650 modified PSA varies a lot with the contact time, and it is 10 times smaller for a short con-
1651 tact than for contact with the duration of 120 seconds. “”

1652 – We thank the reviewer for this question about the minimum contact (preload) time as
1653 it is rather important for perching applications. Please let us answer this in two steps.

1654 First, for ceiling perching, the preload time is approximately 1 s as shown in Fig. [5]
1655 (reproduced below), as well as Figs. [S16], [S23]. For wall perching, the adhesion strength
1656 is more critical and we used the preload time of approximately 3-7 s as seen in Figs. [8]
1657 and [S24], [S31] (Stage II). As you observed, they are notably shorter than 120 s mentioned
1658 in [52].

1659 The reason is, the adhesive used in our rotorcraft is different from the one in [52] in terms
1660 of composition and fabrication. Specifically, our adhesive (a normal PSA) is a simple
1661 copolymer incorporating DOPA and a hydrophobic moiety, whereas the cited adhesive
1662 is a blend of two individual copolymer with temperature-responsive and host-guest moi-
1663 eties. As to the fabrication methods, our adhesive is deposited on a substrate by drop
1664 casting, whereas the cited adhesive is prepared by dip-coating and self-assembly. The
1665 large contrast of adhesion force of the cited adhesive in response to contact time may
1666 attribute to the characteristic two-layer configuration and host-guest linkage between the
1667 two layers.

1668 To avoid a confusion, we revised the manuscript to cite [52] in a more suitable location
1669 (where DOPA-based adhesives are first mentioned in the Introduction).

1670 “Previous studies have identified 3,4-dihydroxyphenyl-L-alanine (DOPA) as the chem-
1671 ical basis for the development of mussel-mimetic polymer [51], [52]. By incorporating
1672 the DOPA moiety into the matrices of PSA, DOPA modified adhesives demonstrate re-
1673 peatable attachment to a variety of surfaces in both dry and wet environments [41], [43],
1674 [44].”

Figure 5: Ceiling perching experiments. (A) Sequential images of the robot perching and taking off from the dry acrylic. (B) The time course of the motor voltages and the power consumption of the robot. Blue dots correspond to the timing of the images in (A). (C) The photographs of from the ceiling perching experiments with different surfaces and conditions.

1675 • *Minor Comment:* “Page 11 Line 150 (When it comes to the wall perching, the incor-
1676 poration of the passive joints serves two purposes: precluding the ceiling adhesive from
1677 sticking to the wall (and vice versa) and assisting the perch up motion towards the wall.):
1678 It may be difficult for a reader to imagine the impact of passive joints because the reader is
1679 not yet acquainted with the working principle of perching that you present later on pages
1680 15-17. Is there a way to put the working principles of ceiling and wall perching (figures
1681 4 and 6) before mechanical design?”

1682 – Thanks for the suggestion. We agree with the reviewer that this sentence may be
1683 difficult to comprehend without a complete picture. In the meantime, we thank it is rather
1684 cumbersome and out of context to explain full perching procedures at this stage. To
1685 resolve this, we expand the respective sentence and make references to Fig. 2D to clarify
1686 these two features.

1687 “When it comes to the wall perching, the incorporation of the passive joints serves two
1688 purposes: precluding the ceiling adhesive from sticking to the wall (and vice versa) and
1689 assisting the *perch up* motion towards the wall. **Nominally, the ceiling adhesive pads**
1690 **are situated on top of the robots, allowing them to readily adhere to the ceiling. With**
1691 **the passive joints, the wall adhesive pads are oriented almost vertically when the robot is**
1692 **flying (see Fig. 2D), permitting them to be in touch with a wall upon approaching. With**
1693 **the revolute joints, the initial contact assists the perch-up maneuver and the designatedsin**
1694 **gap prevents the ceiling adhesive pads from adhering to the wall.”**

1695 • *Minor Comment:* “Page 11, line 159: (To evaluate the adhesion pressure required for the
1696 robot to retain the perched up state. The stress analysis is again considered). English is
1697 awkward: please rewrite. ”

1698 – We thank the reviewer for your thorough evaluation and the suggestion. The full stop
1699 was indeed in a wrong place. We have made an edit accordingly.

1700 “**To evaluate the adhesion pressure required for the robot to remain perched, the stress**
1701 **analysis is again considered.”**

1702 • *Minor Comment:* “Page 11 Line 161 (The analysis provided in Note S5 shows that the
1703 normal adhesion pressure required for the robot to stay attached to the wall is minimal.):
1704 What do you precisely mean by “minimal”? From common sense, one may think that the
1705 normal adhesion force has to be higher or equal to the mass of the drone. ”

1706 – Good advice! This statement should be more precise. In fact, it is the *shear* adhesion
1707 force that directly supports the weight of the robot (not the normal adhesion) in the *wall*
1708 perching. Nevertheless, it is imperative to also analyze the *local* normal adhesion pressure
1709 to ensure that the robot can achieve the moment equilibrium and peel off the adhesive.

1710 To avoid any confusion, we have revised the main text to explicitly differentiate the shear
1711 and normal adhesions as stated below.

1712 “The analysis provided in Note S6 shows that the average normal adhesion pressure re-
1713 quired for the robot to stay attached to the wall is minimal as the weight of the robot
1714 is supported by the shear adhesion. Meanwhile, there exist lower and upper bounds for
1715 the maximum local adhesion pressures for the robot to be in moment equilibrium when
1716 perched and the peel-off to be feasible. With the proposed mechanism, the tenfold differ-
1717 ence between the lower and upper bounds simplifies the synthesis of the adhesive...”

1718 We have also expanded Note S6 to include the calculation that leads to the “tenfold
1719 difference” for completeness

- 1720 • *Minor Comment:* “Page 12 Line 175 (It was previously shown that DOPA modified PSAs
1721 exhibited excellent adhesion both in dry and wet conditions): Are there any limitations of
1722 DOPA modified PSAs? As mentioned earlier, the surface temperature can significantly
1723 change the adhesion properties. Also, there is a relationship between applied adhesive
1724 force (force you apply to stick the DOPA modified PSA onto the substrate) and adhesive
1725 time on the resulting adhesive force.”

1726 – Another good point. Technically speaking, DOPA functionalized PSAs are prone to
1727 oxidation under basic conditions (e.g., pH>7), during which the adhesive dopamine is
1728 converted to non-adhesive o-quinone, resulting unwanted crosslinking and reduced ad-
1729 hesion. As mentioned, temperature is a common influencing factor for PSAs, but PSAs
1730 normally have glass transition temperatures (T_gs) far below room temperature. There-
1731 fore, when temperature is too low (approaching T_gs), the PSAs become too glassy (rigid)
1732 to adhere, whereas when temperature is too high (far above T_gs), the PSAs become too
1733 rubbery (soft) to adhere. Under both conditions, the PSAs exhibit weak or no adhesion.

1734 To acknowledge these limitations, we have included the following sentences in the Dis-
1735 cussion.

1736 “...Unlike dry adhesives that require smooth and dry surfaces [55, 57], the wet adhesive
1737 retains its effectiveness on damp substrates and sticks firmly to non-smooth EVA foam
1738 and wood (Fig. 3 and S5; though, DOPA functionalized PSAs could be susceptible to
1739 oxidation under basic conditions and extreme temperature, i.e., below or far above the
1740 glass transition temperature [51]).”

- 1741 • *Minor Comment:* “Page 12 Line 179 (The adhesive pads were obtained after bonding a
1742 thin p(DMA-co-MEA) layer to a thin laser-cut sheets of PDMS.): It has been shown that
1743 the casting of p(DMA-co-MEA) layer on a pillar structure resulted in a better adhesion
1744 rather than a flat sheet (<https://www.nature.com/articles/nature05968>).

1745 Why have you decided to use a simple layer design instead of using a more benefi-
1746 cial pillar design? Also, different material compositions of DOPA modified PSAs can
1747 have significantly different dry and wet adhesive properties with a specific substrate
1748 (<https://pubs.acs.org/doi/full/10.1021/acsami.9b08429>). In this
1749 work, the researchers compared different material compositions for dry and wet adhesion
1750 for steel and high-density polyethylene. In your work, you tested the adhesion with a
1751 wide range of substrate materials. How did you choose those materials?”

1752 – Indeed, the pillar design would provide higher adhesion than the layer design owing
1753 to the gecko-foot effect. However, the design of our rotorcraft only requires appropriate
1754 or moderate adhesion from the adhesive. Specifically, the adhesion should be sufficiently
1755 enough to ensure the stable perching. Meanwhile, the adhesion should not be too high
1756 to facilitate a take-off. These conditions are described in detail in Notes S5.1 and S6.1.
1757 The adhesion pads employed (43 mm² and 62 mm², Table S2) are already very light and
1758 small. Besides, the adhesion pressure is further increased because of the pillar structure,
1759 smaller adhesive pads would become more sensitive to surface roughness. This will make
1760 the perching less reliable in practice.

1761 To clarify this point, we added an explanation in Material and Methods, under the sub-
1762 section on adhesive fabrication:

1763 “Small PDMS pads for depositing the synthesized polymer adhesive were fabricated from
1764 a thin PDMS sheet (thickness 1 mm), cut to the specific sizes using a CO₂ laser cutter
1765 (Mini 24, Epilog). In this work, the use of flat PDMS bases produced sufficient adhe-
1766 sion pressure, dispensing the need for pillar structures to further boost the adhesion as
1767 evidenced in [41]...”

1768 – Two major considerations for the material selection are (i) representative materials
1769 for real-world robotic applications; and (ii) the differences in surface properties (such as
1770 roughness, porosity, and hydrophilicity).

1771 Regarding the first criterion, four chosen surfaces represent a wide range of materials in
1772 urban and rural settings. Acrylic is similar in appearance to glass. Aluminum represent
1773 metallic structures. EVA foam is used for office wall and ceiling padding. Wood is for
1774 trees.

1775 More scientifically, four surfaces are different in hydrophobicity and roughness. Both
1776 qualities are deemed critical to the adhesion pressure. To demonstrate this in a quantita-
1777 tive fashion, we have added two material characterization tests (Note S4): wettability and
1778 SEM images. The results of the wettability test (Fig. S4) reveal that acrylic, aluminum,
1779 and wood are hydrophilic, whereas EVA foam is hydrophobic. Moreover, the SEM im-
1780 ages verify that all four substrates are vastly distinct in surface morphology (Fig. S5).

1781 Acrylic is extremely smooth. Aluminum has microridge structures that are separated by
1782 1-5 μm . EVA foam is porous and highly non-uniform (pore diameters are 10 to 200 μm).
1783 Wood shows fibrous features at 5-20 μm scale.

1784 To justify the material selection better, we have revised the main text (under **Mussel-**
1785 **Inspired Wet Adhesive** as follows.

1786 “To quantitatively evaluate the suitability of biomimetic wet adhesive for ceiling and wall
1787 perchings, characterization experiments were performed to verify three relevant proper-
1788 ties of the adhesive on four materials: acrylic, aluminum, EVA foam, and wood. These
1789 substrates, representative of common man-made and natural materials, vary widely in
1790 terms of hydrophobicity and roughness as characterized by static contact angles (Fig. S4,
1791 all materials except EVA foam are hydrophilic) and images from a scanning electron mi-
1792 croscope (Fig. S5) acrylic appears distinctly smooth at 1 μm resolution whereas EVA
1793 foam is highly and unevenly porous with the pore diameters on the order of 10-200 μm)
1794 as reported in Note S4. ...”

1795 Please also refer to Note S4, Fig. S4, and Fig. S5 for the accompanying test procedures
1796 and results.

- 1797 • *Minor Comment:* “Page 14 Line 192 (The trend is observed in all four materials tested
1798 (acrylic, aluminum, EVA foam, and wood) in both dry and wet states.): For DOPA modi-
1799 fied PSAs ([https://www.nature.com/articles/s41467-017-02387-2#](https://www.nature.com/articles/s41467-017-02387-2#MOESM1)
1800 [MOESM1](https://www.nature.com/articles/s41467-017-02387-2#MOESM1)), as well as for every other PSA ([https://onlinelibrary.wiley.](https://onlinelibrary.wiley.com/doi/full/10.1002/adem.202200355)
1801 [com/doi/full/10.1002/adem.202200355](https://onlinelibrary.wiley.com/doi/full/10.1002/adem.202200355)), the temperature of the adhesion char-
1802 acterization testing protocol is crucially important as it significantly changes the adhesion
1803 force between the PSA and substrate material. I did not find any information about the
1804 temperature of the interface between PSA and a substrate as well as the temperature of the
1805 environment. Can you provide it and show how these temperatures influence the adhesion
1806 behavior?”

1807 – This is related to the temperature sensitivity issue mentioned earlier. As our PSA is not
1808 designed to be temperature responsive and the applications are not in extreme environ-
1809 nments, we only focus on uses at room temperature. To ensure that this info is not missing
1810 (as the reviewer pointed out), we modified the main text (under **Mussel-Inspired Wet**
1811 **Adhesive**.

1812 “These characterization procedures were conducted with four candidate surface materials,
1813 both under dry and wet conditions at room temperature....”

1814 Please understand that we did not attempt to study the influence of environment temper-
1815 ature on adhesion. However, it is sensible to assume that under low (e.g., 0 °C) and high

1816 (e.g., 40 °C) temperatures, PSAs tend to exhibit reduced or no adhesion as compared to
1817 room temperature.

1818 • *Minor Comment:* “Page 22, lines 291-301: The proposed mechanism is heavier than the
1819 original Crazyflie 2.1. I wonder why it displays longer flight time (455s) compared to
1820 flight time of the unmodified drone (420s).”

1821 – Good question. We strongly believe that the discrepancy is due to the hardware reli-
1822 ability (batteries). This is because the difference in the power consumption is little ($\approx 3\%$)
1823 and small low-cost batteries we used have some variation between units. The difference
1824 in flight times (455 s vs 420) is also insignificant ($\approx 6\%$). In comparison, the recorded
1825 flight time of the original robot (420 s) is also consistent with the official flight time (“up
1826 to 7 minutes”) provided by Bitcraze.

1827 Since the magnitude of the variation 35 s is insignificant when to the perching endurance
1828 (810 s and 1800 s), we believe it does not affect the validity of the results. However, we
1829 still provide an explanation as quoted below.

1830 “The weights of both robots, including markers for the motion capture cameras, were
1831 31.06 g and 32.15 g. The proposed robot consumed marginally more power (8.6 W), on
1832 average, than the original Crazyflie 2.1 (8.3 W) owing to its heavier mass (see Fig. 9A
1833 and Materials and Methods for the measurement methods). **The minute differ-
1834 ence results in comparable flight endurance.** The flight time was recorded as 455 s and
1835 420 s for the proposed and original robots (Fig. 9B). **The small difference in the flight
1836 times is likely due to other variations (such as reliability of the batteries) rather than the
1837 difference in the flight power.”**

1838 • *Minor Comment:* “Page 22, Figure 9C is missing.”

1839 – Well spotted! We have revised the figure with the subfigure C as reproduced below.

Figure 9: Power consumption and operational endurance. (A) Average input power of the robots (i) from hovering flights, (ii) during ceiling perching, (iii) during wall perching, and (iv) during ceiling and wall perchings in endurance flight tests. (B) Total operational times of the robots in extended hovering and perching flights (endurance test). (C) Plots of the power consumed by the robot during the endurance test flights. The timescales in the middle portions, when the robot was perching, are sped up by factors of five and ten.

- 1840 • *Minor Comment:* “Page 25 Line 375 (For future improvement, these shortcomings can be
1841 alleviated through the adjustment on the chemical composition of the adhesive.): DOPA
1842 modified PSAs have different adhesion properties at 25 and 40 °C ([https://www.
1843 nature.com/articles/s41467-017-02387-2#MOESM1](https://www.nature.com/articles/s41467-017-02387-2#MOESM1)). The control of the
1844 surface temperature allows changing the adhesion by 3 times from 0.5 to 3kPa. Why have
1845 you decided to peel off the material instead of using its intrinsic adhesion temperature
1846 dependence? In future work, you may also try to build an adhesive pad with a controlled
1847 interfacial temperature. Such a type of pad can switch the adhesion force from low (at
1848 room temperature) to high (at 40 °C) requiring less energy for landing and taking off.
1849 This improvement can help you to spend even less energy during landing and taking off
1850 reducing the power consumption of the drone.”

1851 – We appreciate the question, particularly the suggestion to use temperature changes for
1852 adhesion control. Though, as we clarified earlier, the cited adhesive is incorporated with
1853 a temperature-responsive moiety, making its adhesion highly dependent on temperature.
1854 This is different from our adhesive.

1855 For this work, the decision not to use the respective adhesive and the temperature con-
1856 trol method is primarily due to the application. The temperature changing process not
1857 only takes time and energy, but also adds extra weight to the small rotorcraft. This will
1858 inevitably decrease the mission time. We have shown that the peeling-based take-off not
1859 only works efficiently but also avoids introduction of additional weight.

1860 Since this is a great suggestion, we have included this potential direction in the **Discussion**
1861 section of the main text.

1862 “... the limitation of the proposed solution remains is the reliance on structured horizontal
1863 and vertical surfaces. This strategy is, therefore, more suitable to an urban usage. **There-**
1864 **fore, the incorporation of temperature-responsive moiety to attain temperature-controlled**
1865 **adhesion [52] could be considered to simplify the takeoff at a cost of increased weight**
1866 **and power.”**

1867 **Reviewer 3**

- 1868 • *Overall Comment:* “The research work proposed is interesting and the methodology
1869 seems sound. Also the work amount to perform the experiments and present the results is
1870 significant. However, I think this paper can be accepted only after minor revisions.

1871 *My remarks and suggestions will mainly concern the vehicle and flight physics.”*

1872 – We would like to take this opportunity to thank the reviewer for professionally assess-
1873 ing our manuscript and for your encouraging comments. We hope you are satisfied with
1874 our revision listed below.

1875 • *Comment:* “Some statements lack of numerical values for example line 215-216 ”ex-
1876 perimental measurements manifest the proximity coefficient of $\gamma = 2.72$, indicating a
1877 significant reduction in the power consumption ”, what is a significant reduction? 2?
1878 10?”

1879 – The original text was indeed a little unclear as pointed out by the reviewer. To clarify
1880 this, we calculated the predicted overall conserved power based on the developed model.
1881 It turns out that the analysis predicts the saving by a factor of 2.2, lower than 2.72. This
1882 is because not all input power is translated into the mechanical power. We have updated
1883 the manuscript accordingly.

1884 “For the developed prototype ($d \approx 2$ mm), experimental measurements manifest the
1885 proximity coefficient of $\gamma = 2.72$. The result suggests a reduction in the overall power
1886 consumption of the robot for the thrust-assisted ceiling perching up to a factor of ≈ 2.2
1887 (calculated based on the analysis in Note S3, the difference is attributed to the power
1888 loss in the motors and electronics). When combined with the support from the adhesive
1889 pads, the overall power consumption during perching can be substantially decreased.”

1890 • *Comment:* “In Figure 4, you mention “the adhesive is preloaded by applying maximum
1891 motor commands”, then “To conserve power, the robot adapts the thrust distribution”.
1892 To conserve the same power as “maximum motor commands”? The objective I guess is
1893 to use the minimum power to stay perched which is contradictory to what I read. More
1894 explanations and reformulation should be given.”

1895 – We thank the reviewer for pointing out confusing sentences related to the ceiling perch-
1896 ing method. This is because the perching process consists of multiple steps and they are
1897 executed sequentially. The maximum command is used only briefly, before lower com-
1898 mands are used when the robot is perched. We have made some clarifications as shown
1899 below.

1900 “A four-stage ceiling perching strategy. Starting below a horizontal overhang, the robot
1901 first ascends to establish contact. Then, the adhesive is preloaded by momentarily apply-
1902 ing maximum motor commands, with propelling forces further amplified by the proximity
1903 effect. Next, to conserve power, the robot adapts the thrust distribution, moving the cen-
1904 ter of collective thrust (depicted as red dots) away from the adhesive pads to lower the
1905 overall force commands. Lastly, takeoff is accomplished through peeling, requiring no
1906 additional mechanism or actuators.”

1907 Furthermore, please find the full formulation of the process, with the quantitative anal-
1908 ysis, described comprehensively in Note S5. The process considers the conditions for
1909 balanced torque and force.

1910
1911
1912
1913
1914
1915
1916
1917
1918
1919
1920
1921
1922
1923
1924
1925
1926
1927
1928
1929
1930
1931
1932
1933
1934
1935
1936
1937
1938
1939
1940
1941
1942
1943
1944

- *Comment:* “In Figure 4, the take off phase is not clear. In the text you mention “thrust-assisted method” and we do not see any thrust in the figure. Is it in the same direction (towards the ceiling) as for contact but lower thrust level? Or in the opposite direction (towards the ground) with low or high thrust? Again this part is missing numerical values. Figure 5 (B) does not answer these questions either. Adding an estimated thrust plot using a simple model with measurements as inputs could be useful.”

– Good point! The term “thrust assisted” in the first submission was not sufficiently explained. The term thrust assistance here refers to the use of small thrust force while the robot is perched. This means the robot consumes small amount of power while perching. However, it eliminates additional actuators and mechanisms for engaging and peeling of the adhesive. The method is, therefore, suitable for small flying robots with severely restricted payload.

To make this clear, we have made the following clarification in the main text, under the section **Ceiling Perching**.

“The strategy developed for the vehicle to repeatedly perch on an overhang makes use of the combined normal adhesion force and the propelling force, reinforced by the proximity effects, to counter the robot’s weight. **With the propellers remain active at low thrust commands when the robot is perched, the peel off is obtained by lowering or powering of the propellers. With the assistance from the propelling thrust, no additional mechanism is required for preloading or disengaging the adhesive pads. In addition to an easy and reliable detachment from the surface, this thrust-assisted method reduces the dependence on the precision of the adhesion pressure, with the power consumption while perching further reduced by the surface-induced aerodynamic interactions.** ”

- *Comment:* “The control strategy or control laws are not presented. I wonder if the UAV is remotely piloted or autonomous? This is an important information to give and develop in the paper.”

– We totally agree, this is an important information that was not stated clearly in the previous version. In this work, the robot operated mostly autonomously. The role of the human pilot is to execute the commands for the robot to begin perching and taking off from the surface. The surface detection, preloading, and thrust tuning, are pre-programmed. In the main results, the robot used both onboard and MOCAP feedback. However, we have also developed a routine for perching with onboard sensors only as detailed in Note **S7**.

Previously, this information is buried in the Supplements and must be deduced by the readers (for example, the perch-up controller is described in Note **S6.4**). In this revision, we have explicitly stated the degree of autonomy and control methods in the main text.

1945 For the ceiling perching, this is

1946 “The proposed four-stage ceiling perching framework (Fig. 4) enables the robot to use
1947 a pair of ceiling adhesive pads to **autonomously and** reversibly perch on an overhang and
1948 conserve the energy over a wide range of substrates. Leveraging onboard feedback, the
1949 maximum adhesion pressure of each particular surface is evaluated on-the-fly to reduce
1950 the perching power. **As a result, a human pilot only needs to (i) initiate the perching
1951 sequence when the robot hovers below the surface; and (ii) instruct the robot to take
1952 off afterwards. During the process, the robot relies primarily on its IMU measurements
1953 for surface detection, command turning, and stabilization using the method described in
1954 Note S5. Position feedback from the motion capture system is used for taking off, but
1955 this can be substituted by the onboard feedback as detailed in Note S7.”**

1956 Please note that Note S5 describes the motor commands (and thrust distribution) used
1957 for the robot when it is perched on the ceiling. This can be considered the control law.
1958 Similarly, we have revised the text explaining the wall perching process in the main text
1959 to be more comprehensive. Though, the full details, including the perch-up controller,
1960 can be found in Note S6.

1961 “Perching on the wall leverages shear adhesion to support the robot weight and signif-
1962 icantly reduce the collective thrust to achieve equilibrium of moments and power con-
1963 servation. The developed strategy, elaborated in Fig. 6 and Note S6, renders the wall
1964 perching operation reliable by allowing the robot to momentarily apply substantial com-
1965 pressive preload to the adhesive, taking into account the actuation limit and the design
1966 and configuration of the robot. Once completely perched, the robot retains its stability
1967 with minimal propelling thrust, markedly lowering the power consumption compared to
1968 the regular hovering flight. **Similar to the ceiling perching, the developed wall perch-
1969 ing method and control law allow the robot to perch autonomously. The motion capture
1970 system used for the takeoff can be replaced by onboard sensors as described in Note S7.”**

1971 In addition to the points listed above, we made minor revisions and corrections throughout.
1972 We remain extremely enthusiastic about this work. We hope you agree that our work contributes
1973 to the advancement of aerial robots.

1974 Sincerely,

1975 Zuankai Wang

1976 Pakpong Chirarattananon

Reviewers' comments:

Reviewer #1 (Remarks to the Author):

Unfortunately, after reading the rebuttal letter, I am still not convinced if the contributions are enough to consider this paper for publication. Below, I provide the reasons.

Line 1242 (of the rebuttal letter): The authors claim that added actuation payload is negligible in their design, a main advantage not reported before. This claim is not correct. The same claim can be made with almost all works that use adhesive fibers, a celebrated example is the work by Murphy et al., IJRR, 2011. Note that in Murphy's work, actuation is for locomotion.

Line 1247: The authors say that the ceiling (ground) effect has never been used for perching. In fact, a plethora of works already uses this phenomenon to not only perch on ceilings and walls but also traverse (a more complex problem than perching) on these surfaces (see -- just as one example -- VertiGo from Disney Research Zurich).

Line 1254: Authors claim the main novelty is not in their adhesive developments (see Line 1351 in the rebuttal letter), but in how they combine adhesion with proximity effects for perching. If we exclude the adhesive pads used in this draft, how is the draft distinguishable from the work by H. Tsukagoshi et al., ICRA, 2015? In Tsukagoshi's work, the thrust vectoring and suction cups are employed to perch in identically the same way the authors use thrust vectoring and their adhesive pads. In fact, in Tsukagoshi's work the suction cups and the proximity effect are used together not only for perching but also manipulation purposes.

Line 1264: Authors claim only a few robots can perch on the ceiling (seven is the number they provide). It is not seven. This list can be extended with many works on robots that can even traverse ceilings. Second, assuming only seven works successfully did what is reported in this draft, this means this work is not novel and has no new messages for its readers.

Line 1476: The authors mention that the work's main contribution is not the wet adhesive. Then, this statement leads to confusion about the main contribution of this work because the reader can easily assume that the work presented here enables wet adhesion, something that has remained relatively unexplored as opposed to dry adhesion. Otherwise, impressive dry adhesion on ceilings was reported a long time ago (see IEEE Spectrum article by M. Pope, 2016).

Overall, I do not believe this paper has new messages for its readers.

Reviewer #2 (Remarks to the Author):

The authors have done a remarkable job in addressing all our comments and those of the other reviewers. The manuscript now clearly identifies the novelty claims and backs them up with convincing data. We find the combination of ceiling effect and adhesive pads innovative and worth of publication in Communications Engineering.

Dario Floreano and Yegor Piskarev

PS: For sake of completeness, it may be interesting to add to the comparative table in the supplementary material our previous work on drone perching on flat surfaces with dry adhesives, but do not feel obliged to do so!

L. Daler; A. Klaptocz; A. Briod; M. Sitti; D. Floreano
A Perching Mechanism for Flying Robots Using a Fibre-Based Adhesive

In Proceedings of ICRA 13, Karlsruhe, May 6-7, 2013.

Reviewer #3 (Remarks to the Author):

The authors have answered my comments adequately. Hence, I do not have further comments or questions.

Response to Comments

For clarity, the provided comments are in blue. Our revised texts, taken directly from the revised manuscript, are shown in red. If there is anything further we can do to clarify any of these points, please let us know.

Reviewer 1

- *Overall Comment:* “Unfortunately, after reading the rebuttal letter, I am still not convinced if the contributions are enough to consider this paper for publication. Below, I provide the reasons.”

– We appreciate the reviewer’s thoughtful consideration of our responses and the opportunity to further clarify our contributions.

Our work makes the following key contributions:

- 1) A lightweight perching mechanism for micro drones that dispenses with additional actuators by leveraging the ceiling effect. This enables flight capabilities to be maintained for vehicles with strict weight budgets.
- 2) The characterization and quantification of the ceiling effect on a micro drone in perching applications. This effect has been observed but not systematically implemented for perching. Our measurements verify the actual benefits in real flight.
- 3) A novel combination of the ceiling effect and wet adhesive for perching. This combination expands the perching abilities of small drones to more surfaces and conditions while minimizing mass increase.

We believe these specific contributions can advance the state of the art of micro drone perching and are worth considering for publication. However, we are open to clarifying or modifying our responses based on your remaining concerns. Please let us know how we can address them.

- *Major Comment 1:* “The authors claim that added actuation payload is negligible in their design, a main advantage not reported before. This claim is not correct. The same claim can be made with almost all works that use adhesive fibers, a celebrated example is the work by Murphy et al., IJRR, 2011. Note that in Murphy’s work, actuation is for locomotion.”

– We appreciate the reviewer bringing up a specific example, which allows us to understand your viewpoint. However, there are several fundamental distinctions between *perching* and *climbing*. Our claims only apply to perching robots, not climbing robots, since they present inherently different challenges and requirements.

The cited work (Murphy et al.) develops a *climbing* robot that uses dry adhesive, but its mechanism cannot be directly applied to perching drones. The added mass of its servomotor-driven foot pads is acceptable for climbing but would prevent micro drones from lifting off. This highlights how perching small flying robots is uniquely difficult due to strict weight limits.

Perching refers to flight-capable agents, which are exclusively about flying robots or animals. This implies our literature review is comprehensive for perching drones. While being exhaustive is impossible, we did not miss major work. Our claim that "the added payload is negligible in our design, a main advantage not reported before" thus stands.

"The manuscript now clearly identifies the novelty claims and backs them up with convincing data. We find the combination of ceiling effect and adhesive pads innovative and worth of publication in *Communications Engineering*."

– To address the reviewer's concern, we reinforce the introduction to mention the importance of added mass on aerial vehicles and state how this aspect is unique to small flying robots and different from climbing robots not capable of flying.

".. These additional components account for an appreciable portion of the final vehicle mass. As illustrated in Fig. S9, the perching mechanisms (including added actuators) constitute over 15% of the total mass for vehicles under 100 g. While added mass generally poses less of a challenge for terrestrial locomotion. The aerial domain's reliance on generating sufficient lift and the strong dependence of power on mass renders the mass budget for flying vehicles, especially small drones, far more stringent. For multirotor platforms, momentum theory predicts the scaling between the aerodynamic power thrust T of a spinning propeller as $P_a \sim T^{3/2}$ [40, 41], implying that a 15% increase in weight, for instance, nominally leads to a 23% rise in power consumption."

– In summary, while added mass is less critical for climbing robots, it poses a significant challenge for perching aerial vehicles with strict weight budgets. Therefore, our claim that the added actuation payload is negligible in our perching drone design, which enables flight capabilities to be maintained, remains valid and distinct from prior work on climbing robots.

- *Major Comment 2*: "The authors state that the ceiling (ground) effect has never been used for perching. However, many works already use this phenomenon to not only perch on ceilings and walls but also traverse on these surfaces, such as VertiGo from Disney Research Zurich."

– We thank the reviewer for referring us to another climbing robot. We are able to access a short documentation from: <https://la.disneyresearch.com/publicat>

ion/vertigo/.

To clarify, the cited climbing robot VertiGo:

- Is a wall-climbing robot, not a perching aerial vehicle
- Lacks evidence of leveraging the ceiling effect or quantifying its benefit

While two propellers are present, and their role is to apply thrust toward the wall, there is no evidence, measurements, or any mention of the ceiling effect. There is certainly no characterization or measured contribution of the ceiling effect. In fact, the ceiling effect may not be present at all if the distance is not close enough.

In our view, it is not sufficiently scientific to conclude that the mentioned robot leverages the ceiling effect. Even if it does, the benefit is never quantified. Please allow us to provide another analogy; even if one researcher observes and reports the propeller-surface interaction but does not systematically characterize the degree of the force or the benefit, it should also not preclude other researchers from publishing the characterization results in a technical manner.

In conclusion, while we appreciate the reviewer’s reference to the VertiGo climbing robot, we believe that it is important to clarify that this is a wall-climbing robot, not a perching robot. There is no evidence or mention of the ceiling or proximity effect.

– To resolve this concern, we highlight our original claim: The ceiling effect allows us to use the propelling thrust to preload the wet adhesive directly, instead of relying on spring-loaded mechanisms as elastic energy storage. This is feasible as the robot takes advantage of the aerodynamic effect induced by a nearby surface.

“to preload the wet adhesive, the propelling thrust is used directly (instead of the use of elastic energy stored in a mechanism as seen in [15, 18, 23-26, 28, 31, 32, 34, 37]). This is feasible as the robot takes advantage of the aerodynamic effect induced by a nearby surface.”

– In conclusion, while we appreciate the reviewer’s reference, the key distinctions show that the cited climbing robot does not dilute the novelty of our lightweight and efficient perching strategy for small aerial vehicles.

- *Major Comment 3:* “The authors claim that the main novelty is not in their adhesive developments but in how they combine adhesion with proximity effects for perching. Excluding the adhesive pads used in this draft, how is the draft distinguishable from the work by H. Tsukagoshi et al., ICRA, 2015? In Tsukagoshi’s work, the thrust vectoring and suction cups are employed to perch in identically the same way the authors use thrust vectoring and their adhesive pads. In fact, in Tsukagoshi’s work, the suction cups and the proximity effect are used together not only for perching but also manipulation purposes.”

– We thank the reviewer’s request for clarification. To clarify, there are two key issues:

- 1) The novelty of this work lies in the robotics application of the ceiling effect and adhesive, not the adhesive development itself. This is similar to how the novelty of Waalbot II (Murphy et al., *IJRR*, 2011) is in the climbing mechanism using dry adhesive, not the adhesive development.

- 2) Our work is technically, visually, and substantially different from Tsukagoshi et al., *ICRA 2015*:

- Different working principles: ceiling effect vs. vacuum suction
- Different implementations: 30-g flying robot vs. 1,700-g robot with pumps and pressure gauges
- Different abilities: perching on overhangs and walls vs. walls only

Quantitatively, the mass difference is evident in Fig. S9.

Figure S9: Mass ratio of the perching mechanisms (refer to Table S1 for itemized data and sources). The plot displays the mass of the mechanisms against the total mass of the robots in logarithmic scales, categorized by the perching ability. The dashed lines represent the weight ratios of the mechanisms with respect to the total mass.

Regarding the concern, excluding the adhesive would still not make the works indistinguishable. Different adhesives call for different requirements and tailored mechanisms. The problem is aggravated for limited-payload aerial vehicles. This manuscript solves the issue of perching such small drones.

In conclusion, while we appreciate the reviewer’s request, the key distinctions show that our work’s novelty lies in the lightweight and low-power robotics application of the ceiling effect and adhesive for perching micro drones.

- *Major Comment 4:* “The authors claim that only a few robots can perch on the ceiling (seven is the number they provide). However, this claim is not accurate as there are many works on robots that can even traverse ceilings. Assuming only seven works have successfully done what is reported in this draft, this means that this work is not novel and has no new messages for its readers.”

– We appreciate the viewpoint. However, we would like to also address this concern in three steps.

To clarify, we stated that out of 22 robots in Table S1, only seven can perch on a wall *or* ceiling, with three on the ceiling. Perching on the ceiling is not trivial.

Among 22 (now 23) works cited in S1, each has strengths and weaknesses; novelty can lie in specific strengths. For example, Roderick et al. 2021’s grippers accommodating high-speed collisions is a novelty, despite other branch-perching drones.

In conclusion, while we appreciate the reviewer’s comment, our work’s novelty lies in the specific strengths of low-mass low-power ceiling/wall perching without additional actuators, not being the first ceiling-perching drone.

- *Major Comment 5:* “Line 1476: The authors mention that the work’s main contribution is not the wet adhesive. Then, this statement leads to confusion about the main contribution of this work because the reader can easily assume that the work presented here enables wet adhesion, something that has remained relatively unexplored as opposed to dry adhesion. Otherwise, impressive dry adhesion on ceilings was reported a long time ago (see IEEE Spectrum article by M. Pope, 2016).”

– We appreciate your view that the use of wet adhesive is relatively unexplored. We do agree on this point.

– To clarify, our statement that the main contribution is not the wet adhesive development means we report the mechanism and the use of wet adhesive for perching, not its development. This distinction is important. We have to be careful on this point not to overstate the contribution of the work. We believe it is the same for many robotics papers that cannot claim that their main contribution is the development of *dry adhesive*.

– Regarding the IEEE Spectrum article by Pope *et al.*, 2016 (accessible at <https://spectrum.ieee.org/microspines-make-it-easy-for-drones-to-perch-on-walls-and-ceilings>), we are fully aware of this state-of-the-art

robot. It was cited and discussed in the manuscript as [34], which is the actual publication in IEEE Transactions on Robotics (not an online article). In this work and the online article,

- uses *microspines* (previously used for perching by Kovac et al., 2009), not dry adhesive, for perching and *climbing*.
- has heavier mass (11 g mechanism, 37 g total) than our solution (Fig. S9)
- lacks use of proximity effect to reduce power consumption
- cannot perch on smooth surfaces

In conclusion, our work’s novelty lies in using wet adhesive for perching with low mass and proximity effect use, not adhesive development.

Reviewer 2

- *Overall Comment:* “The authors have done a remarkable job in addressing all our comments and those of the other reviewers. The manuscript now clearly identifies the novelty claims and backs them up with convincing data. We find the combination of ceiling effect and adhesive pads innovative and worth of publication in Communications Engineering.

Dario Floreano and Yegor Piskarev”

– Thank you for your positive feedback on our revised manuscript. We appreciate your recognition of our efforts in addressing the comments and feedback from both you and the other reviewers. We are glad that our manuscript now clearly identifies the novelty claims and provides convincing data to support them. We are pleased to hear that you, as a highly recognized researcher, find our work innovative and worthy of publication.

- *Minor Comment:* “PS: For sake of completeness, it may be interesting to add to the comparative table in the supplementary material our previous work on drone perching on flat surfaces with dry adhesives, but do not feel obliged to do so!

L. Daler; A. Klaptocz; A. Briod; M. Sitti; D. Floreano, A Perching Mechanism for Flying Robots Using a Fibre-Based Adhesive, In Proceedings of ICRA 13, Karlsruhe, May 6-7, 2013. ”

– Thank you for your suggestion to include your previous work in the comparative table in the supplementary material. We agree that this would provide additional context and completeness to the table, and we have included your work in the revised manuscript.

Reviewer 3

- *Overall Comment:* “The authors have answered my comments adequately. Hence, I do not have further comments or questions.”
 - We would like to once again express our gratitude for your constructive feedback. Your comments have been very helpful in improving our manuscript.

In addition to the points listed above, we made minor revisions and corrections throughout. We remain extremely enthusiastic about this work. We hope you agree that our work contributes to the advancement of aerial robots.

Sincerely,

Yufeng Kevin Chen
Zuankai Wang
Pakpong Chirarattananon